

# Soil carbon stock estimates in a nationwide inventory: evaluating performance of the ROMUL and Yasso07 models

Aleksi Lehtonen[1], Tapio Linkosalo[1,2], Mikko Peltoniemi[1], Risto Sievänen[1], Raisa Mäkipää[1], Pekka Tamminen[1], Maija Salemaa[1], Tiina Nieminen[1], Boris Ťupek[1], Juha Heikkinen[1] and Alexander Komarov[3]

[1]Natural Resources Institute Finland (Luke), PO Box 18, FI-01301 Vantaa, Finland
[2]University of Helsinki, Department of Forest Sciences, PO Box 27, FI-00014 Helsinki, Finland
[3]Late from Institute of Physicochemical and Biological Problems in Soil Science of the Russian Academy of Sciences, 142290 Institutskaya ul., 2, Pushchino, Moscow Region, Russian Federation

*Correspondence to*: Aleksi Lehtonen (aleksi.lehtonen@luke.fi)

**Abstract.** We test whether litter quality, litter quantity and weather data are enough to estimate soil carbon stocks by models. We also test whether inclusion of soil water holding capacity improves soil carbon stock model estimates. Litter input was estimated from stem volume maps provided by the National Forest Inventory, while understorey vegetation was estimated using new biomass models. The litter production rates of trees were based on previous research, while for understorey biomass those were estimated from measured data. We applied Yasso07 and ROMUL models across Finland and ran those models into steady state; thereafter, measured soil carbon stocks were compared with model estimates. We found that the role of understorey litter input is underestimated when the Yasso07 model is parameterised, especially in northern Finland. We also found that the inclusion of soil water holding capacity in the ROMUL model improved predictions, especially in southern Finland. Our results imply that the ecosystem modelling community and greenhouse gas inventories should improve understorey litter estimation in the northern latitudes. Our simulations and measurements show that models using only litter quality, litter quantity and weather data underestimate soil carbon stock in southern Finland and this underestimation is due to omission of the impact of droughts to the decomposition of organic layers.

## 1 Introduction

Soil carbon is a significant component of terrestrial carbon stocks and understanding its dynamics under changing climate is crucial. However, the significance of different drivers of soil carbon stocks is still unknown. If we want to understand the role of different abiotic and environmental factors to soil carbon stocks and their dynamics, we have to combine experimental research with process-based models. On the other hand, we can establish soil carbon inventories in order to quantify soil carbon stocks and their change. Conventional soil inventories measuring various nutrients, carbon contents, bulk densities and stoniness (e.g. Gamfeldt et al. 2013) allow us to study the distribution of soil carbon across landscapes and correlations between different soil properties. Generally, it is shown that soil carbon inventories are able to produce soil





maps and covariates between soil carbon quantities and other variables, such as various nutrients; although, the sample size of these inventories is usually not adequate for national-level soil carbon stock change assessment, with few exceptions (e.g., Sweden and Germany).

According to various international climate agreements, countries are requested to report annual carbon stock changes of soils under different land-uses and under land-use change. The majority of countries apply soil carbon models, like Yasso07 (Tuomi et al. 2011) and CENTURY (Parton et al. 1987) to estimate soil carbon stock changes. The scientific community is also aiming to predict future soil–climate change feedbacks on a global level using Earth system models (ESMs). The ESMs are tested against soil carbon measurements in order to evaluate model performance but unfortunately results have been poor. Guenet et al. (2013) present a test whereby soil carbon stocks predicted by ORCHIDEE model were plotted at a plot level against measurements and failed to display any correlation. Similarly, Todd-Brown et al. (2013) concluded that most ESMs are not able reproduce measured soil carbon stocks at a grid level.

Individual soil carbon models are tested against repeated soil inventories and it is found that models are able to estimate soil carbon stock change of the same magnitude as was measured. The limitation of this conclusion was that uncertainties of both measurements and model estimates are often higher that actual estimates (Ortiz et al. 2013, Rantakari et al. 2012). While the uncertainties between model output and real measurements can establish whether models agree with data or not, they put less emphasis on whether all the most important soil carbon stock drivers were included in these models.

Simplistic soil carbon models like Yasso07 (Tuomi et al. 2011) are driven only by weather conditions and by litter input; while more complex models, like ROMUL (Chertov et al. 2001) also include the nitrogen cycle and the impact of soil water holding capacity on decomposition (Linkosalo et al. 2013). It is clear that soil properties affect soil carbon stocks (Schimel et al. 1994, Six et al. 2002) and therefore they are explicitly included in complex models. For example, in the CENTURY clay content limits decomposition (Parton et al. 1987) and due to the low specific surface area of clay minerals in the model, clay rich soils have larger passive soil carbon stocks and lower C:N ratios (Parfitt et al. 1997). In simplistic models soil properties are omitted. For example, in Yasso07 the role of soil properties on soil carbon stock accumulation is included only implicitly through the model calibration with large datasets (Tuomi et al. 2011). Although the simpler models lack some predominant drivers of soil carbon accumulation, the strength of these models lies in their easier calibration with data; however, the impact of soil properties, especially nutrient status, on the accuracy of estimated soil carbon stock estimates needs to be re-evaluated in both CENTURY and Yasso07 models (Tupek et al. 2016).

It is well known that decomposition and soil respiration is controlled by water content, whereby in dry soils lack of water slows down decomposition, while excess water reduces it by limiting oxygen diffusion (Skopp et al. 1990). For boreal forest conditions, Pumpanen et al. (2003) proposed a model whereby maximum soil respiration drops after the relative water content of soil reaches a level of 60%. The limiting effect of soil moisture on decomposition in dry or water-saturated soils is widely included in models, although the degree of this dependency varies widely between models (Sierra et al. 2015).

In addition to the model structure, precise and accurate estimates of litter input quantity and quality are also essential for successful model applications. Stand-alone soil models rely on forest inventory data or other external estimates for getting



correct litter inputs, while ecosystem models utilise plant sub-models to describe vegetation productivity and litter inputs. A common feature of ecosystem models and stand-alone soil models is that often understorey vegetation is neglected during the calibration and application of models, resulting in biased litter inputs. This omission is more critical in boreal landscapes where the contribution of understorey vegetation increases with northerly latitude. For example, Yuan et al. (2014) report

that the bryophyte biomass contributes 20%−60% of the total normalized difference vegetation index (NDVI) in high northern latitudes.

In order to assess the necessary drivers for soil carbon models in Finland we tested the performance of Yasso07 and ROMUL models against measured nation-wide data. The Yasso07 and ROMUL models differed in their time steps (annually versus daily), determination of the litter quality (litter solubility versus nitrogen content and litter source) and complexity of

10 drivers (ROMUL including the impact of nitrogen and soil water holding capacity to decomposition). Specifically with these models we tested:

1. Whether the litter quantity, litter quality and weather data are enough to estimate spatial trends of carbon stocks in the upland soils of Finland. We hypothesise that by accounting for soil properties, ROMUL would outperform Yasso07 and that improving the estimation of understory litter input would positively affect accuracy of predicted soil carbon stocks.

2. Whether variation in soil water holding capacity affects carbon stocks through drought limitation on decomposition. We hypothesise that an increased fraction of coarser soil textures, like sand, increases soil carbon stocks through increased drainage and reduced decomposition.

We use Yasso07 and ROMUL soil models to estimate steady-state carbon stocks for upland soils in Finland on a spatial $10 \times 10$ km$^2$ grid. We run Yasso07 model with parameters based on Scandinavian data (Rantakari et al. 2012) and also with

20 parameters based on global data (Tuomi et al. 2011). The parameterisation for the ROMUL model was the same as in the original publication (Chertov et al. 2001). Yasso07 and ROMUL models run with identically estimated litter quantity, quality and climate data. In addition, the ROMUL model was driven with constant and varying soil water holding capacity based on digital soil maps of Lilja and Nevalainen (2006) as in Linkosalo et al. (2013). Furthermore, we develop new models for the understorey litter input and apply them alongside soil carbon models. Simulated soil carbon stocks are evaluated

against measured soil carbon stocks.

## 2 Material and methods

Soil carbon simulations were performed for $10 \times 10$ km$^2$ grid across Finland. This grid is used for meteorological data

prediction (Venäläinen et al. 2005). Litter input was estimated for the same grid. From the grid, only locations that were on upland soils and on forest, according to Food and Agriculture Organization of the United Nations (FAO) forest definitions were chosen. This classification is based on Multisource National Forest Inventory products (Tomppo et al. 2008).



## 2.1 Tree biomass

Firstly, stem volume maps by tree species from the National Forest Inventory 9 (NFI9, 1996-2003) were used, according to Tomppo et al. (2011), to account for large-scale variation of stem volume across Finland. These variations are primarily driven by soil properties, climate, site productivity, forest management techniques and tree species distribution.

Secondly, biomass expansion factors (BEFs, $Mg/m^3$) were estimated for main tree species groups, namely: Scots pine (*Pinus sylvestris*), Norway spruce (*Picea abies*) and for broadleaved species (mainly *Betula* sp.). These BEFs were estimated for each biomass component (foliage, branches, bark, stemwood, stump and woody roots). Biomasses were estimated for sample trees in NFI10 based biomass models by Repola (2008, 2009); thereafter, mean BEFs were estimated at a cluster level (a cluster is formed of 10 to 15 field plots) by dividing the sum of given biomass components by the estimated sum of stem

volumes. Biomass estimates for trees were based on biomass models, where diameter at breast height, tree height, crown height, increment of five years and bark thickness are used as predictors. To upscale BEFs across Finland, we applied collocated co-kriging (Bivand et al. 2008) by species group, to account for large-scale spatial correlation and co-variation of tree allometry. We used the *gstat* (Pebesma 2004) package of R (R Core Team 2014) for estimation. For Scots pine and Norway spruce we removed linear trends of latitude and longitude (using uniform coordinate system of Finland, YKJ), while

for deciduous trees only trends of latitude were removed. For all species groups and components we assumed spherical variogram functions. For the details of the used biomass models see Appendix 7c by Statistics Finland (2014). Biomass components for each grid point were obtained by multiplying stem volume maps by species with component species-specific BEF estimates that were estimated via co-kriging for the same grid.

Biomass of harvest residues and natural mortality were estimated based on forest statistics and NFI data (Ylitalo 2013).

From statistics, we attained an estimate of the stem volume of annual loggings and natural mortality by region (forestry centres) and these were subsequently converted to biomass with BEFs. These BEFs were based on a subset of permanent sample plots of NFI9 (1996–2003) and NFI10 (2004–2008). BEFs for logging were estimated separately from the subset of logged plots and these logging specific BEFs were used in the estimation of biomass of harvest residues. Furthermore, energy use of stumps and harvest residues were deducted from regional soil inputs, based on regional wood energy use

(Ylitalo 2013). For biomass of natural mortality, BEFs were estimated based on data from those trees that died on permanent sample plots between measurements. Thereafter the volume of natural mortality was multiplied with corresponding BEFs. This procedure followed principles of Finnish greenhouse gas (GHG) inventory (Statistics Finland 2014).

Fine root biomass was estimated based on the work of Lehtonen et al. (2016). We selected a simple model formulation, where a natural logarithm of fine root mass was estimated as a function of the natural logarithm of stem volume. See Model

1 in Lehtonen et al. (2016) for details. This model was used to approximate general fine root mass levels as a function of stem volume for each $10 \times 10 \ km^2$ grid point.





## 2.2 Understorey vegetation biomass

In order to estimate the litter input of understorey vegetation to soils, we developed models for vegetation biomass. The relationship between the visually estimated percentage cover of plant species including vascular plants, bryophytes and

lichens (projected onto the forest floor using $30 \times 30$ cm$^2$ frames) and their living biomass was studied in 18 forest plots across Finland (Table 1). The plots are part of the International Co-operative Programme on Assessment and Monitoring of Air Pollution Effects on Forests (ICP) intensive monitoring plot network (e.g. Merilä et al. 2014). Altogether 28 systematically selected biomass samples were taken once from each plot during the period 2002 to 2009. Vascular plants were divided into aboveground (shoots) and belowground (rhizomes and roots in organic layer) parts. The study was carried

out at the time of maximum biomass and vegetation growth between the end of July and end of August. Half of the plots were located in the north (five in *Pinus sylvestris*, three in *Picea abies* and one in *Betula pubescens* dominated stands) and the other half were in the south (four in *Pinus sylvestris*, four in *Picea abies* and one in *Betula pendula* dominated stands). The site types of the plots ranged from poor to rich fertility level (Table 1).

In order to predict the vegetation biomass, we built linear mixed models based on the relative coverage of five functional

plant groups (dwarf shrubs, grasses, herbs, bryophytes and lichens). The biomass of understorey vegetation was estimated using models that correlate vegetation coverage with measured biomass. These models were estimated for the five aforementioned main species groups and for their belowground parts, if applicable. These models were estimated separately for southern and northern Finland. We used a linear mixed model with plot-level random effects using the *lme* command in *nlme* package (Pinheiro et al. 2012) of the R environment for the estimation (R Core Team 2014).

Each model was weighted according to the land area of different site types in southern and northern Finland using the weights option in *lme*. This was done to ensure that the sample of understorey biomass plots did not underestimate the weight of the most common site types (those of medium fertility). The weighting was performed on land areas based on NFI11 data (Ylitalo 2013). Models were linearised by taking natural logarithms from biomass and coverage. For model parameters, bias correction and their uncertainties see Table 3.

To quantify the biomass of understorey vegetation we used species-specific coverage measurements of permanent sample plots from 1995 forest inventory data (Mäkipää and Heikkinen 2003). We applied collocated co-kriging methods to account for the correlation between species groups and thereafter we generalised measured understorey coverage on a $10 \times 10$ km$^2$ grid on upland soils across Finland (Bivand et al. 2008). The co-kriging method was similar to that of BEF estimation described above. First, we de-trended all species group coverage (shrubs, herbs and grasses, lichens and bryophytes) by

linear trends of latitude and longitude (using uniform coordinate system of Finland, YKJ). After that we estimated variograms and cross-variograms and applied co-kriging methods to predict coverage for understorey coverage for Finland by *gstat* package of R (R Core Team 2014). For all species groups we assumed spherical variograms with common range of 100 km.





### 2.3 Litter input estimation

To estimate litter input from living biomass components to the soil, litter turnover rates were used (Table 2). For needles, we used the detailed $10 \times 10$ km$^2$ grid litter turnover rates reported by Tupek et al. (2015). For tree fine roots we assumed that

the life span was 1.18 years for southern Finland and 2 years for northern Finland (Kleja et al. 2008, Leppälammi-Kujansuu et al. 2014). In order to interpolate fine root turnover rates for Finland we assumed a linear dependency for turnover rate and mean temperature sum (mean for 1981–2010). For areas with a temperature sum (5 C° threshold) higher than 1200 degrees a turnover rate of 85% was used and for areas where temperature sums less than 700 degrees a rate of 50% was used. Turnover rates were interpolated between 700 and 1200 degrees (see Supplement).

For branch and coarse roots litter, the constant turnover rates of the branches were used across Finland (Lehtonen et al. 2004, Muukkonen and Lehtonen 2004). For bark litter we assumed a constant ratio to stem biomass according to Viro (1955) and Mälkönen (1977). For fellings and natural mortality we assumed that all biomass was left to the site, excluding stems for wood used and harvesting residues for energy use. The quantity of wood use was based on regional forestry statistics (Ylitalo 2013).

The turnover rates of different functional plant groups were partly based on literature (Table 2) but we also estimated some rates from our own biomass samples of understorey vegetation described above. We calculated the proportion of the current-year growth out of the total living biomass of dwarf shrub shoots to estimate the turnover rates for aboveground plant parts. We used the average values of deciduous and evergreen dwarf shrubs. Most of the grasses growing in boreal forests (like *Deschampsia flexuosa*) are perennial and we used the ratio between the living and dead biomass to estimate the turnover rate

of those species. The aboveground parts of the most herb species (like Maiathemum bifolium) are annual, so we used 100% for their turnover rate. Furthermore, we used the number of annual growth segments in the green upper parts of the bryophytes to estimate their rate. We assumed that over a long time period the estimate for annual biomass growth corresponds to the amount of litter input to the soil from understorey vegetation.

### 2.4 Application of soil carbon models

Carbon stocks were estimated by running Yasso07 and ROMUL models into a steady state (i.e., a state where carbon input for the model equals carbon flux due to decomposition). For a dynamic model a steady state is a state to which model aims with given inputs. Here model inputs were: average climatic conditions (1961–2012) and litter inputs for each grid point, depending on dominant tree species and understorey vegetation coverage plus regional estimates of litter from harvestings

and natural mortality (sc. forestry centres). If we assume that this average level of inputs and climate has remained steady over centuries, then our soils should approach steady state conditions. Litter inputs were given to these models as $10 \times 10$ km$^2$ grid points and weather data of that given grid point was then used for model input.



When evaluating model results for Yasso07, fresh woody litter and recently dead wood litter were excluded from model state variables, while all non-woody material and humus boxes (including more heavily decomposed material from fine-woody- and coarse-woody litter) were accounted as soil carbon stock. This separation was done as Yasso07 slows down decomposition of logs of high diameter, which increases soil steady state carbon stock estimates. Data from Biosoil does not

include dead wood masses in soil carbon stocks. For ROMUL we included all model state variables to the comparison, noting that log size does not affect decomposition.

### 2.4.1 Climate data

Daily weather data for steady state simulations are available as kriging estimates since 1961 at a $10 \times 10$ km$^2$ resolution across Finland (Venäläinen et al. 2005). Weather data for the grid points on upland soils and on forested land, according to

10 multisource NFI, were included (Tomppo et al. 2008).

Daily weather predictions for a $10 \times 10$ km$^2$ grid were aggregated for the Yasso07 model from 1961 to 2012. The average temperature, temperature amplitude and precipitation of each grid cell were provided to Yasso07 to estimate the impact of climate on soil carbon stock steady states. The ROMUL model was driven by mean daily temperature and precipitation for each grid cell.

### 2.4.2 Yasso07

The Yasso07 soil carbon model (Tuomi et al. 2011) is driven by litter quantity, litter quality and weather (temperature and precipitation). This model is widely used in the greenhouse gas inventories of European countries (e.g., Finland, Norway and Switzerland). This model has been also successfully coupled with climate-carbon cycle model ECHAM5/JSBACH (Thum et al. 2011).

The Yasso07 is a simple dynamic model with fluxes and state variables. The model has five compartments: acid, water, ethanol, non-soluble and humus boxes. Organic matter flows between these boxes and to atmosphere variably according to weather conditions (Fig. 1). The model builds on the assumption that organic matter solubility defines litter quality, while decomposition rates while fluxes are estimated numerically, without *a priori* assumptions. The model was calibrated using a large database of litter and wood decomposition measurements, and measurements of age chronosequences of soil carbon

stocks (Liski et al. 2005, Liski et al. 1998, Rantakari et al. 2012, Tuomi et al. 2011). The Yasso07 soil carbon model parameter values were estimated with Markov chain Monte Carlo methods for which the source code is publicly available through the model website (www.syke.fi) (Tuomi et al. 2011). Here, we used the Yasso07 model with Scandinavian parameters (Rantakari et al. 2012) and with global parameters, these being different from Tuomi et al., (2011) parameters. Yasso07 was run with total litter input (from trees and understorey vegetation) and also with litter excluding that of

understorey vegetation. The model was applied with annual time-steps. See the Supplement for more details of Yasso07.



### 2.4.3 ROMUL

ROMUL is a soil organic matter decomposition model developed by Chertov et al., (2001). It describes the flux of organic matter through the decomposition process, separated in cohorts of different litter origins: leaves, shoots, trunks, coarse roots, fine roots and ground vegetation. Litter from above ground (leaves, shoots and stems) falls onto the forest floor, whereas the

5 root litter decomposes in the mineral soil layer. In the final stages, all decomposing matter ends up in a common storage of semi-stable humus residing in the mineral soil layer (Fig. 1). Decomposition rates of each cohort depend on the nitrogen and ash content of the specific litter type and for all cohorts the soil temperature and water content modify the decomposition rate. We applied the ROMUL model using daily time-steps of the environmental variables impacting the decomposition. The ROMUL model explicitly simulates the flux of nitrogen through the decomposition process and therefore produces estimates

for mineralised N in addition to the soil organic matter and N storages and $CO_2$ release from the soil. All model parameters, except the decomposition rate, which is dependent on soil water content, were taken from the Chertov et al. (2001) paper.
In this paper we adopt decomposition rate functions depending on soil water content and the model for soil water dynamics as described in Linkosalo et al. (2013). We applied the ROMUL model with soil water holding capacity data, obtained from digital soil maps of Finland (Lilja and Nevalainen 2006). Total soil water holding capacity data was extracted to a $10 \times 10$

$km^2$ grid, when available. To evaluate the impact of soil water holding capacity variability, we also repeated the ROMUL simulation assuming a constant soil water holding capacity for the whole simulation area. As the ROMUL model segregates the organic and mineral soil layers, we assumed that 20% of the total soil water holding capacity was in the organic layer and 80% in mineral soil layer. See the Supplement for more details of ROMUL, the source code and parameter estimates.

**2.4 Biosoil – soil carbon measurements**

The Biosoil dataset resulted from EU-funded Forest Focus monitoring program and includes measurements of soil carbon and various other ecosystem variables from 2006. Data consists of 521 sample plots located across Finland. These plots were established as a subset of 3009 permanent sample plots from 1985–86 (see Mäkipää and Heikkinen 2003).
Biosoil sample plots have radii of 11.28 m (400 $m^2$). Trees and understorey vegetation coverage were measured. Soil carbon

samples were taken separately from the litter and humus layers and mineral soil layers at 0–10, 11–20, 21–40 and 41–80 cm depth. In total 10 or 20 subsamples were taken from the organic layers with a cylinder (diameter = 60 mm) and 10 tube (diameter = 23 mm). Alternatviely, 5 spade subsamples were taken from the 0–10 cm mineral soil layer, a further 5 spade subsamples were taken from the layers 11–20 and 21–40 cm and one subsample from the layer 41–80 cm. Viro (1952) rod penetration method was used to quantify the volume of stones and boulders in the surface soil layer. Several properties were

analysed from soil samples in the laboratory: soil texture, cation exchange capacity (CEC), base saturation (BS), pH, organic carbon and total nitrogen concentrations. Aqua regia extractable P, Ca, K, Mg, Mn, Pb, Na, Ni, Fe and S were also measured



(Derome et al. 2007). Soil carbon stocks were estimated down to 1 m depth by extrapolation of 40–80 cm measurements for the 80–100 cm layer.

Measured soil carbon stocks were grouped into different latitudinal bands forming a gradient in Finland to which we compared soil carbon estimates from Yasso07 and ROMUL models against measured data. Comparisons were made with two different widths of latitude bands: firstly we used bands of 100 km and thereafter bands of 20 km. Bands were included in the analysis when there were four or more Biosoil observations in a band. We also identified those Biosoil plots that had been under the Litorina Sea 7000–8000 BP (Miettinen 2004, Sohlenius et al. 1996) using map products produced by the Geological Survey of Finland (Eronen 1974).

We also studied differences between mean stand and soil properties for two regions in southern Finland (Region 1 with a latitude below 60˚46', and Region 2 with a latitude between 60˚46' and 62˚34').

## 3 Results

Aboveground biomass of dwarf shrub vegetation was twice as large in the North compared to the South (Fig. 2) and the difference was even more pronounced with belowground biomass (Fig. 3). For bryophytes we found large variability, especially when their ground coverage approached 100%; therefore, our model is valid up to 80% ground coverage and for ground coverage over 80% we estimate mean biomass of bryophytes to be 1124 kg ha$^{-1}$ and 1055 kg ha$^{-1}$, respectively, for southern and northern Finland. All understorey biomass models displayed substantial unexplained variation, owing to the fact that we had only ground coverage [%] as an explanatory variable and because we pooled together different species into these functional groups (e.g., dwarf shrubs with different leaf weights, like *Vaccinium myrtillus* and *Empetrum nigrum*). See Table 3 for model parameter values and their uncertainties.

The amount of litterfall from trees and understorey vegetation showed opposite trends with latitude. Total litter input of trees and other vegetation in southern Finland was four times higher than that in northern Finland. Though, the understorey vegetation at northern sites displayed much higher litter inputs than those in southern Finland. According to our results, fall of understorey litter in eastern and northern Lapland was equal to that from trees (Fig. 4). We estimated mean litterfall from understorey vegetation to approximately equate to 473 kg ha$^{-1}$ and 863 kg ha$^{-1}$ of carbon for southern and northern Finland, respectively. In the Finnish GHG inventory these estimates, based on Muukkonen et al. (2006), were 506 kg ha$^{-1}$ and 665 kg ha$^{-1}$ for southern and northern Finland, respectively. Our estimate of mean litter input from trees was 1962 kg ha$^{-1}$ and 903 kg ha$^{-1}$ of carbon for southern and northern Finland, respectively.

The soil carbon stock estimates by Yasso07 model using global parameterisation (Tuomi et al. 2011) were systematically larger than measured data across Finland, whereas those obtained with Scandinavian parameterisation (Rantakari et al. 2012) were in the same magnitude as the Biosoil data (Fig. 4A and B). In addition to the realistic level of soil carbon stock, the model that was based on parameterisation with the Scandinavian data also reproduced decreasing soil carbon stock trend from south to north, as displayed in the Biosoil data. Excluding understorey litter from the model input improved the match



between Yasso07 soil carbon stocks simulated using global parameters and Biosoil data, whereas for Yasso07 using Scandinavian parameters, the same exclusion resulted in underestimation of soil carbon stocks, especially in northern Finland (Fig. 4C and D). The ROMUL model predictions generally agreed with Biosoil data when soil water holding capacity was taken into account. The inclusion of soil water holding capacity in ROMUL introduced high variation in soil

carbon stocks between dry and moist grid points in southern Finland (Fig. 4E). When the ROMUL model was driven with constant soil water holding capacity, it was unable to reproduce decreasing soil carbon stocks across Finland and the model underestimated carbon stocks, especially in the south (Fig. 4E and F). Large deviations between the data and the model estimates were also seen for the largest Biosoil soil carbon stocks of the southernmost plots (Fig 3).

The ROMUL model using the soil water holding capacity was the only model able to reproduce the increase in measured soil

carbon stocks at the southernmost plots. All other models predicted a substantial decrease in the soil carbon stocks for southern region, which was not observed in measurements (Fig. 4). Soil properties of southernmost plots (Region 1 (R1) with a latitude below 60˚46') were different compared to soil properties of forests further north (Region 2 (R2) with a latitude between 60˚46' and 62˚34'). The southern coast had lower silt content and higher sand content, but simultaneously higher carbon stocks in the organic layer than the other region (Table 4). Also understorey vegetation differed between these

regions and the southern coast had lower *sphagnum* species and herbs species coverage, indicating that soils there were drier. We also tested site type distributions between R1 and R2 but the p-value from a chi-square test was > 0.4, indicating that site fertilities did not differ. This is also supported by the measured C:N ratios, which did not differ between these regions (Table 4). These southern sites were also younger and 29% of plots were under the Littorina Sea ~7000–8000 years BP, while the share of younger soils was only 6% for the slightly more northern region.

We found that Yasso07 model applications excluding litter input from understorey vegetation had the best agreement with the observed latitudinal gradient when evaluating with one-to-one plots (Fig. 6). The modelled and measured soil carbon stock fits were poor for the Yasso07 application with global parameters, including the understorey vegetation litter input and for the ROMUL application not using soil water holding capacity data. These model applications showed no correlation with measurements and also failed to map the south to north soil carbon stock decrease (Fig. 6). The lowest root mean square

error was obtained with the ROMUL model applied with soil water holding capacity data.

## 4 Discussion

We tested whether litter quality, litter quantity and mean climate are sufficient for estimating spatial trends in soil carbon stocks and if soil texture with low water holding capacity introduces water limitation to decomposition. While testing our

hypothesis we found that Yasso07 and ROMUL models were able to predict carbon stocks of the same magnitude as that of measurements for Finland; however, these models experienced more challenges when their performance was evaluated for smaller regions, like the southern coast of Finland.

We also found that litter input from understorey vegetation is equal of that from trees in northern Finland. This emphasises the large role of understorey vegetation in compensating trees in an ecosystem carbon cycle, especially under light and



nutrient limited growth conditions. To sum up, litter input from understorey vegetation has to be quantified properly when soil carbon models are parameterised. The results of our biomass models for the understorey agreed with previous estimates for southern Finland but for northern Finland our estimates were substantially larger. Yasso07 was parameterised using understorey litter inputs ranging from 40–60 g carbon $m^{-2}$ $a^{-1}$ and therefore Yasso07, with updated understorey litter estimates, overestimated the level of soil carbon stocks. Our data for understorey vegetation biomass was mostly from stands that were over 60 years old and therefore more data for younger sites are needed. Furthermore, the litter input of belowground understorey vegetation was uncertain due to limited data on the life span of roots. More research is therefore needed to confirm the high contribution of dwarf shrub vegetation to total belowground litter input.

When evaluating modelled and measured soil carbon stocks against northern latitude we could see that two out of six simulations failed to map measured soil carbon stock decreases towards the north (Fig 4). Five model simulations showed that southern coast had less carbon than the next region further north, while the Biosoil data showed the opposite. Only the ROMUL model, using soil water holding capacity data from Lilja and Nevalainen (2006) and applied as in Linkosalo et al. (2013), was able to estimate the largest soil carbon stocks for southern Finland, similar to Biosoil measurements (Fig. 5). This indicated that litter quality, litter quantity and climate data are not sufficient when estimating spatial trends of soil carbon stocks. When we evaluated Biosoil data for these regions, the southern sites were better drained and drier that those in North. Better drainage was indicated by higher sand content, significantly lower silt content and significantly lower *Sphagnum* and herb vegetation coverage (Table 4). We also found out that the southern coast experiences a 1.3 C˚ higher mean annual temperature than the next region to the north. Therefore, the organic layer was larger for southern coast likely due to limited decomposition during dry spells (Table 4). This result indicated that the quantity of precipitation alone was not a sufficient modifier for decomposition but when complemented with soil water holding capacity results improved. This finding supports the use of models including soil texture and water holding capacity.

According to our results, the time step of the simulation plays a critical role; running models with monthly and annual time steps excludes extreme conditions and may produce biased estimates of carbon stocks simply due to fact that non-linear models are run with mean conditions (Dalsgaard et al. 2016). This is especially critical with soil moisture, which has a bell shaped relation to decomposition (Sierra et al. 2015, Skopp et al. 1990). Compared to daily time steps of ROMUL, model simulations with longer time steps (e.g., a year as in Yasso07) exclude both extreme dry and extreme moist conditions, leading to underestimation of steady states soil carbon stocks. On the other hand, reduction of the decomposition rate during limited and excess water conditions are not well known. Sierra et al. (2015) compared the effect of moisture on decomposition in different models with the largest variability for dry and saturated conditions. They showed that a moisture index of 0.2 can have a decomposition modifier between 0.2 and 1 and similarly with a moisture index of 1 this modifier can be anywhere between 0 and 1, depending on the model. These large discrepancies indicate differences between soil properties of sites used for model calibration and between conceptualisations of soil moisture.

The Yasso07 and ROMUL model simulations with litter input and climate data alone were not enough to reproduce the observed soil carbon stocks. Model application without soil moisture impacts on decomposition failed under conditions



where soil drainage played a significant role in limiting decomposition. The use of water holding capacity was critical for accurate soil carbon stock estimation by the ROMUL model, while Yasso07 performed best when the variation in litter input correlated with that of the observed soil carbon stocks (Fig. 6). The fact that Yasso07 soil carbon stocks were more accurate when litter input of understory vegetation was omitted from simulations suggests that the model calibration did not account

for the whole range of understory litter input. In order to improve model performance, shorter time steps complemented with more detailed topography and soil properties are needed to map the impact of extreme events to soil carbon decomposition (e.g., droughts and water logging conditions). However, three model simulations out of six produced relatively accurate estimates of soil carbon stocks compared to the measurement means of smaller regions (Fig. 6). This suggests that improved calibration with updated understorey litter and accounting for the soil properties as with ROMUL (using data on water

holding capacity) produces model estimates that agree regionally with data.

Our findings confirm the fact that GHG inventory methods and soil modules of Earth system models need to be improved by incorporating the impact of soil texture and soil moisture to decomposition. This is a prerequisite for unbiased soil carbon stock and stock change estimates.

### Code availability

The fortran code of the ROMUL is given in the supplement, while fortran code for Yasso07 is available through website [http://code.google.com/p/yasso07ui/], for details see Supplement.

### Acknowledgements

We thank the researchers and supportive staff that contributed to the data used in this study: Biosoil project and the National Forest Inventory (NFI) project. We also thank Jari Liski and his team for developing and supporting the Yasso07 soil model.

We also acknowledge our funders, namely the Ministry of Forestry and Agriculture and the Ministry of the Environment – Finland (project ''Developing methods for GHG-inventory'' and ''GHG inventory of Finland''). We thank the staff of the Natural Resources Institute Finland, LUKE for handling and analysing the biomass samples of plant species from the ICP Forests Level II plots during the EU/Life+ FutMon programme. We also acknowledge EU DG ENV and especially Life09 ENV/FI/000571 Climforisk and Life12 ENV/FI/00409 Monimet for funding.

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



Table 1. Site description of ICP Forests Level II plots studied for understorey vegetation biomass.

| Plot no | Area | Region | North coord. | East coord. | Tree species | Site type | Forest type | Stand age | Basal area m² | DF | GR | HE | BR | LI | Year of sampling | n above/below |
|---|---|---|---|---|---|---|---|---|---|---|---|---|---|---|---|---|
| 1 | Sevettijärvi | 1 | 7723 | 3573 | 1 | 5 | UVET | 210 | 13.52 | x | - | - | x | x | 2009 | 28/12 |
| 2 | Pallasjärvi | 1 | 7543 | 3377 | 1 | 4 | EMT | 100 | 17.85 | x | - | - | x | x | 2003 | 28/28 |
| 3 | Pallasjärvi | 1 | 7549 | 3384 | 2 | 3 | HMT | 150 | 15.42 | x | x | x | x | - | 2003 | 28/28 |
| 4 | Sodankylä | 1 | 7472 | 3485 | 1 | 4 | EMT | 80 | 19.66 | x | - | - | x | x | 2003 | 28/28 |
| 5 | Kivalo | 1 | 7360 | 3484 | 2 | 3 | HMT | 80 | 17.95 | x | x | - | x | - | 2002 | 28/28 |
| 6 | Kivalo | 1 | 7364 | 3488 | 1 | 4 | EMT | 65 | 18.94 | x | - | - | x | x | 2002 | 28/28 |
| 32 | Kivalo | 1 | 7371 | 3486 | 3 | 3 | HMT | 58 | 14.39 | a | a | a | x | - | 2009 | 28/0 |
| 21 | Oulanka | 1 | 7359 | 3612 | 2 | 3 | HMT | 180 | 21.03 | a | a | a | x | - | 2009 | 28/0 |
| 20 | Lieksa | 2 | 7012 | 3687 | 1 | 4 | EVT | 140 | 22.27 | a | - | a | x | - | 2009 | 28/0 |
| 10 | Juupajoki | 2 | 6866 | 3353 | 1 | 4 | VT | 90 | 23.55 | x | x | x | x | - | 2002 | 28/28 |
| 11 | Juupajoki | 2 | 6863 | 3359 | 2 | 2 | OMT | 90 | 30.4 | x | x | x | x | - | 2002 | 28/28 |
| 12 | Tammela | 2 | 6730 | 3325 | 2 | 3 | MT | 70 | 33.08 | x | x | x | x | - | 2002 | 28/28 |
| 13 | Tammela | 2 | 6727 | 3330 | 1 | 4 | VT | 70 | 29.27 | x | x | x | x | - | 2002 | 28/28 |
| 16 | Punkaharju | 2 | 6854 | 3627 | 1 | 4 | VT | 90 | 31.95 | x | - | x | x | - | 2002 | 28/28 |
| 17 | Punkaharju | 2 | 6858 | 3622 | 2 | 2 | OMT | 80 | 30.76 | - | - | x | x | - | 2002 | 28/28 |
| 33 | Punkaharju | 2 | 6862 | 3622 | 3 | 1 | OMaT | 27 | 16.13 | - | a | a | - | - | 2009 | 28/0 |
| 34 | Luumäki | 2 | 6763 | 3515 | 1 | 5 | CT | 60 | 13.55 | x | - | - | x | x | 2009 | 28/12 |
| 35 | Luumäki | 2 | 6756 | 3513 | 2 | 3 | MT | 60 | 28.21 | x | x | x | x | - | 2009 | 28/12 |

Region (1 = northern Finland, 2 = southern Finland); tree species (1 = Scots pine, 2 = Norway spruce and 3 = deciduous trees); site type (1–5, from rich to poor fertility level); forest site types (abbreviations explained in Salemaa et al. (2008), Table 1); plant species groups in biomass samples: DF = dwarf shrubs, GR = grasses, HE = herbs, BR = bryophytes and LI = lichens. x = biomass sample includes both aboveground and belowground (in organic soil layer) part of vegetation, a = sample includes only aboveground vegetation, - = plant group does not grow in the plot. n = number of samples (each sized 30 x 30 cm²) for aboveground and belowground biomass.



Table 2. Litter turnover rates for tree and understorey biomass by component, species and region.

| Biomass | Species | Region | Value [%] | Reference |
|---|---|---|---|---|
| Leaves | Scots pine | | 27% [18-34%] | Tupek et al., (2015) |
| Leaves | Norway spruce | | 13% [9-15%] | Tupek et al., (2015) |
| Leaves | Broadleaved | | 79% | Tupek et al., (2015) |
| Branches | Scots pine | | 2% | Lehtonen et al. (2004) |
| Branches | Norway spruce | | 1.25% | Muukkonen and Lehtonen (2004) |
| Branches | Broadleaved | | 1.35% | Lehtonen et al. (2004) |
| Bark | Scots pine | | 0.3% | (Mälkönen 1977, Viro 1955) |
| Bark | Norway spruce | - | - | (Mälkönen 1977, Viro 1955) |
| Bark | Broadleaved | | 0.01% | (Mälkönen 1977, Viro 1955) |
| Coarse roots | Scots pine | | 2% | Lehtonen et al. (2004) |
| Coarse roots | Norway spruce | | 1.25% | Muukkonen and Lehtonen (2004) |
| Coarse roots | Broadleaved | | 1.35% | Lehtonen et al. (2004) |
| Fine roots | Scots pine | South | 85% | Kleja et al., (2008) |
| Fine roots | Norway spruce | South | 85% | Kleja et al., (2008) |
| Fine roots | Broadleaved | South | 85% | Kleja et al., (2008) |
| Fine roots | Scots pine | North | 50% | Leppälammi-Kujansuu et al., (2014) |
| Fine roots | Norway spruce | North | 50% | Leppälammi-Kujansuu et al., (2014) |
| Fine roots | Broadleaved | North | 50% | Leppälammi-Kujansuu et al., (2014) |
| | | | | |
| Aboveground | Dwarf shrubs | - | 37% | This study |
| Belowground | Dwarf shrubs | - | 8% | This study& Helmisaari et al. (2015) |
| Aboveground | Grasses | - | 33% | This study |
| Belowground | Grasses | - | 59% | Leppälammi-Kujansuu et al., (2014) |
| Aboveground | Herbs | - | 100% | |
| Belowground | Herbs | - | 59% | This study & Leppälammi-Kujansuu et al., (2014) |
| Aboveground | Bryophytes | - | 42% | This study |
| Aboveground | Lichen | - | 10% | Kumpula et al., (2000) |





Table 3. Parameter estimates for understorey models by species groups and regions. Where $\beta_0$ is the intercept and $\beta_1$ is a slope of the fixed part of the model and bc is the bias correction. Var $b_0$, Cov $b_0 b_1$ and Var $b_1$ originate from variance-covariance matrix of the fixed part of the model. Var pop lists unexplained variance and Var plt lists unexplained variance after plot as a random effect.

| Group | Com. | Reg. | $\beta_0$ | $\beta_1$ | bc | Var$\beta_0$ | Cov$\beta_0 \beta_1$ | Var$\beta_1$ | Var Pop | Var Plt |
|---|---|---|---|---|---|---|---|---|---|---|
| Dwarf shrub | Ab | NF | 3.62 | 0.948 | 0.135 | 0.115 | -0.028 | 0.008 | 0.285 | 0.136 |
| Dwarf shrub | Ab | SF | 2.653 | 1.107 | 0.186 | 0.081 | -0.015 | 0.005 | 0.45 | 0.16 |
| Grass | Ab | - | 2.75 | 0.918 | 0.292 | 0.129 | -0.03 | 0.01 | 0.688 | 0.273 |
| Herb | Ab | - | 1.116 | 1.08 | 0.445 | 0.074 | -0.018 | 0.008 | 0.832 | 0.572 |
| Bryophytes | Ab | - | 3.13 | 0.795 | 0.281 | 0.301 | -0.072 | 0.018 | 0.633 | 0.306 |
| Lichen | Ab | - | 3.69 | 0.894 | 0.109 | 0.047 | -0.012 | 0.005 | 0.27 | 0.175 |
| Shrub | Bl | NF | 6.278 | 0.45 | 0.161 | 0.086 | -0.02 | 0.006 | 0.315 | 0.241 |
| Shrub | Bl | SF | 3.73 | 0.831 | 0.458 | 0.15 | -0.035 | 0.011 | 1.06 | 0.867 |
| Grass | Bl | NF | 3.368 | 0.589 | 0.639 | 0.169 | -0.039 | 0.017 | 1.423 | 0.859 |
| Herb | Bl | SF | 1.877 | 0.906 | 0.709 | 0.27 | -0.067 | 0.03 | 1.195 | 0.898 |

Com. = compartment: Ab = aboveground, Bl = belowground

Reg. = region: NF = northern Finland, SF = southern Finland



Table 4. Mean soil and forest stand properties for R1 and R2 and 1.96 times their standard error of mean (SEM). R1 is south from 60˚46', while R2 lies between 60˚46' and 62˚34'northern latitude. P-values based on two sided t-test, where group variances are assumed to be different variables where p-value < 0.05 are in bold. Sample sizes were 31 and 127 plots, for southern (R1) and northern (R2) areas, respectively. For mean temperature and precipitation sample sizes were more than 150 based on Finnish Meteorological Institute (FMI) weather grid (Venäläinen et al. 2005).

|  | Sand [%] | Silt [%] | Clay [%] | BS O [%] | BS M [%] | CEC O [cmol(+)/kg] | CEC M [cmol(+)/kg] | pH (H2O) | C:N O | C:N M | Sphagnum [%] | Lichen [%] | Herb [%] | G [m2] | Decid [%] | Spruce [%] | Lito [%] |
|---|---|---|---|---|---|---|---|---|---|---|---|---|---|---|---|---|---|
| R1 mean | 64.92 | **30.08** | 4.98 | 67.2 | 23.26 | 32.23 | 5.27 | 4.23 | 26.41 | 23.11 | **0.3** | 0.86 | **4.43** | 23.45 | 15.55 | 39.81 | **29** |
| R1 sem | 7.31 | **5.39** | 2.64 | 6.08 | 5.5 | 3.22 | 0.54 | 0.12 | 1.25 | 1.57 | **0.53** | 1.05 | **1.83** | 3.29 | 6.87 | 12.01 | **16** |
| R2 mean | 57.76 | **37.09** | 5.15 | 68.8 | 27.72 | 29.06 | 4.91 | 4.17 | 25.13 | 21.6 | **1.51** | 0.45 | **8.56** | 23.9 | 19.28 | 37.1 | **6** |
| R2 sem | 3.34 | **2.55** | 1.2 | 2.29 | 3.28 | 0.87 | 0.43 | 0.05 | 1.56 | 0.79 | **0.98** | 0.26 | **2.18** | 1.43 | 4.58 | 6.09 | **4** |
| p-value (t.test) | 0.09 | **0.03** | 0.91 | 0.63 | 0.18 | 0.07 | 0.31 | 0.4 | 0.21 | 0.1 | **0.04** | 0.48 | **0.01** | 0.81 | 0.38 | 0.69 | **0.01** |
|  | T mean [C] | Prec [mm] | extrCd [mg/kg] | extrPb [mg/kg] | extrK [mg/kg] | extrP [mg/kg] | extrNa [mg/kg] | extrNi [mg/kg] | extrCa [mg/kg] | extrMg [mg/kg] | extrFe [mg/kg] | extrS [mg/kg] | extrMn [mg/kg] | TWI | Altitude [dm] | SOC O [Mg ha-1] | SOC M [Mg ha-1] |
| R1 mean | **4.91** | **580.71** | **0.48** | **47.88** | 1149.16 | 943.61 | 139.71 | 10.8 | 4638.39 | 992.52 | 5157.74 | 1435.39 | 459.47 | 6.97 | **588** | **25.82** | 31.65 |
| R1 sem | **0.09** | **6.25** | **0.05** | **3.62** | 138.59 | 80.16 | 14.87 | 1.03 | 1044.39 | 329.89 | 984.63 | 85.95 | 167.12 | 1.83 | **124** | **5** | 7.15 |
| R2 mean | **3.62** | **545.66** | **0.38** | **36.02** | 1225.03 | 940.32 | 131.19 | 11.07 | 4058.02 | 1063.51 | 4675.6 | 1380.96 | 537.84 | 8.79 | **1119** | **17.46** | 35.23 |
| R2 sem | **0.05** | **2.28** | **0.03** | **5.77** | 85.23 | 38.99 | 11.11 | 1.05 | 256.21 | 159.48 | 643.98 | 55.04 | 148.54 | 1.14 | **55** | **1.57** | 3.31 |
| p-value (t.test) | **0** | **0** | **0** | **0** | 0.37 | 0.94 | 0.38 | 0.73 | 0.3 | 0.71 | 0.43 | 0.31 | 0.5 | 0.1 | **0** | **0** | 0.38 |

Here sand, silt and clay content are given as a percentage. Base saturation (BS). cation exchange capacity (CEC) and C:N ratios are given separately for organic layer (O) and for mineral soil layer (M). pH is measured in water. Abundance of ground vegetation given as percentage coverage. Basal area of trees (G) and relative shares of deciduous species and Norway spruce from basal area. Lito indicates the proportions of sites that were under the Littorina Sea 8000–7000 BP. Aqua regia extractable K, P, Na, Ni, Ca, Mg, Fe, S and Mn unit mg/kg. Topographical wetness index (TWI). Altitude of plots based on 10 m resolution map layer. Amount of carbon in organic and mineral soil horizon.





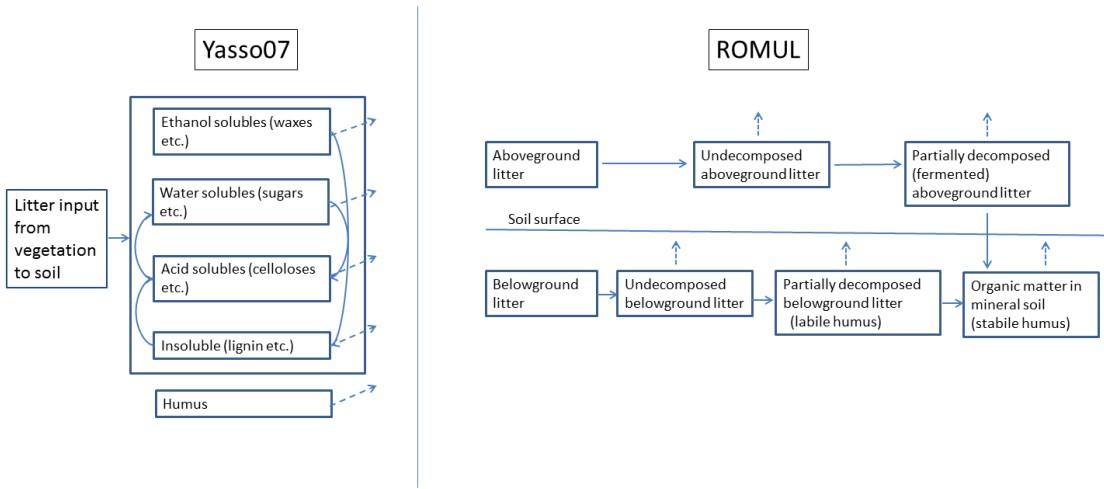

**Figure 1: Schematic illustration of Yasso07 (left) and ROMUL (right) soil carbon models. Solid arrows indicate fluxes of organic matter, while dashed arrows indicate CO$_2$ fluxes to atmosphere.**





**Figure 2: Models and data points for aboveground understorey biomass (A = Dwarf shrubs in northern Finland, B = Dwarf shrubs in southern Finland, C = Grasses, D = Herbs, E = Mosses and F = Lichen). Solid line based on modelling that takes into account site fertility distribution for Finland by weighting, while dashed line is the estimate without weights.**






**Figure 3. Models and data points for belowground understorey biomass (A = Dwarf shrubs in northern Finland, B = Dwarf shrubs in southern Finland, C = Grasses and D = Herbs). Solid line based on modelling that takes into account site fertility distribution for Finland by weighting, while dashed line is the estimate without weights.**



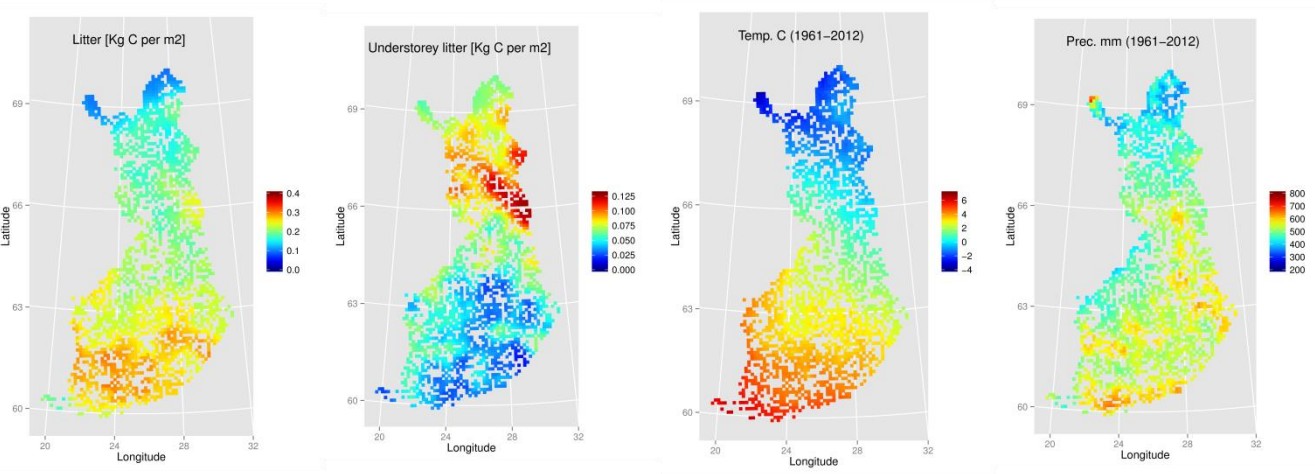

**Figure 4.** Maps for total litter input [Mg ha$^{-1}$], understorey litter input [Mg ha$^{-1}$], mean annual temperature [C°] and mean annual precipitation [mm].





**Figure 5. Latitudinal trends of measured and modelled soil carbon stocks across Finland. X-axis is the North coordinate according to Finnish YKJ system and Y-axis is the soil carbon stock Mg ha⁻¹. Grey dots are individual model estimates and red line is a 2ⁿᵈ order *loess* fit. Panel A is Yasso07 with Rantakari et al. (2012) parameters, while B is with Tuomi et al. (2011) parameters. C and D are same as A and B, but without understorey litter input. E and F are with ROMUL model where E includes soil water holding**




**capacity data and F is with constant soil water holding capacity. Black dots are means from Biosoil data for each latitude band and whiskers are 1.96 times standard error of the mean.**



**Figure 6. One-by-one plots for mean model estimates and mean Biosoil measurements of soil carbon stocks for 41 latitudinal bands across Finland. Panel A is Yasso07 with Rantakari et al. (2012) parameters, while B is with Tuomi et al. (2011) parameters. C and D are same as A and B but without understorey litter input. E and F are with ROMUL model where E is with soil water holding**





capacity data and F is with constant soil water holding capacity. $R^2$, slope of the regression and the slope standard error are reported. RMSE is based on the difference between model estimate and measurements.