# Peer review of "Forest soil carbon stock estimates in a nationwide inventory: evaluating performance of the ROMULv and Yasso07 models in Finland"

_Geoscientific Model Development, 2016_

## Short Comment (SC1) · 11 Aug 2016

Dear authors,

In my role as Executive editor of GMD, I would like to bring to your attention our Editorial version 1.1:

http://www.geosci-model-dev.net/8/3487/2015/gmd-8-3487-2015.html

This highlights some requirements of papers published in GMD, which is also available on the GMD website in the 'Manuscript Types' section:

http://www.geoscientific-model-development.net/submission/manuscript_types.html

In particular, please note that for your paper, the following requirement has not been met in the Discussions paper:

- "The main paper must give the model name and version number (or other unique identifier) in the title."

The above statement is part if the requirements for the model description papers. However, model evaluation papers are closely linked to model description papers and therefore it would be beneficial to specify the exact version of the models the evaluation was performed with in the title. Please consider this upon revision of your article.

Yours,

Astrid Kerkweg

––––––––––––––––––––––––––––––

---

## Referee Comment (RC1) · Anonymous Referee #1 · 15 Sep 2016

This is an interesting study that tested the performance of two forest soil carbon models for their ability to predict SOC stocks across Finland. Specifically, the authors investigated the explanatory power of the parameters litter quality and quantity as well as understorey vegetation and water holding capacity. The strength of the paper is the detailed description of model parameters, particularly the estimation of litter input and understorey vegetation biomass, which are difficult to detect. Although some of the these estimations might be afflicted with uncertainty (for example, a spatial extrapolation of understorey vegetation by kriging for total Finland based on only 18 plots is hardly meaningful), I think that a rough estimation of an important but difficult parameter is better than not account for it at all. This is particularly true when the main

objective is to compare model performance rather than a precise SOC prediction.

There is only one important point which is not clear so far, the depth of the SOC estimates. I did not find any information about the depth for which SOC was predicted. From the biosoil data that was extrapolated to 1 m depth it seems that this was the depth also the models predicted SOC. However, due to the fact that C dynamics in subsoils are fundamentally different from topsoil dynamics, I can hardly imagine that such simple soil models are able to predict also subsoil SOC stocks sufficiently. From my point of view the structure of those models only allows a prediction for topsoils (0-20 cm), for deeper parts stabilization mechanisms of mineral-associated SOC must be taken into account. At least, model performance should be tested not for the total depth but for different depth increments (e.g. 0-20, 20-40 cm etc.).

After a revision in terms of this issue and following minor points the paper is acceptable for publication.

Title: I would include "Forest soil carbon" and "Finland"

P1, L11: generally, the manuscript is well-written, but the first sentence of the abstract is not very good ("We test...weather data are enough..."), please rephrase.

P2, L3: References for Sweden and Germany?

P2, L11: to reproduce

P2, L17: here is becomes clear for the first time that the paper is about forest SOC, however, this could be clarified from the beginning with a clear focus on forest SOC models and references to similar studies.

P2, L20: CENTURY model

P5, L31: the kriging approach was only shortly mentioned. Due to its importance the overall performance of this important step should be described in more detail, e.g. by showing variograms.

---

## Referee Comment (RC2) · Anonymous Referee #2 · 16 Sep 2016

Being able to predict the current inventory is very different than being able to accurately assess changes in carbon stocks due to land use or climate. The authors more or less make this point in the introductory comments but then go on to try to model the current inventory data as a means for improving soil carbon models. Given the detailed inventory data already available, a spatial model seems to be the way to go to improve the national inventory. If the authors are trying to develop a model that can predict year-to-year variations in soil carbon stocks then calibrating models on a few sites with really good long-term data seems more powerful than trying to recreate mean latitudinal trends on soil carbon. Additionally, most of the year-to-year change in soil carbon is going to come from land use and management decisions which have essentially been

ignored here because this MS only focuses on established forests on upland soils.

I really struggled with this paper. It seems that two separate ideas on model testing ending up getting merged together in this MS along with some new measurements on understory biomass and litter turnover. Given the goals of the paper, I would have thought that one model needed to be used that could have different levels of complexity added or removed. Given the fundamental differences between the Yasso07 and RO-MUL models I do not see how it is possible to test the hypothesis that "accounting for soil properties" would improve model performance and then go on to say that time step (annual v. daily) might matter. The second hypothesis is only related to the ROMUL model. Since most of the hypothesis testing relates to the ROMUL model, why not just try to add a better litter module onto the ROMUL model and see if this matters?

Model success was never really defined. The author's suggest that the ROMUL model with some information on soil water holding capacity is the superior model but based better representation of southern soil carbon data but the goodness-of-fit statistics presented in Fig 6 are equivocal on this point.

Detailed comments are included in annotated PDF file.

Please also note the supplement to this comment:
http://www.geosci-model-dev-discuss.net/gmd-2016-144/gmd-2016-144-RC2-supplement.pdf

―――――――――――――――――――――

[Figure]

**Supplement:**

[revised manuscript text omitted]

---

## Author Comment (AC1) · 2 Nov 2016

Response to referees for paper:

" Forest soil carbon stock estimates in a nationwide inventory: evaluating performance of the ROMULv and Yasso07 models in Finland" by *Lehtonen et al.*  *Submitted for publication in Geoscientific Model Development journal.*

Dear Editor,

We acknowledge constructive comments and editorial remarks provided to our manuscript. We find comments useful and addressed them as described below (our reply is given in *italics*), note that page and line numbers refer to version with track changes done:

**Comments by Executive Editor:**

1. "The main paper must give the model name and version number (or other unique identifier) in the title."

The above statement is part if the requirements for the model description papers. However, model evaluation papers are closely linked to model description papers and therefore it would be beneficial to specify the exact version of the models the evaluation was performed with in the title. Please consider this upon revision of your article.

*This was corrected accordingly and ROMUL is now ROMULv, where "v" refers to decomposition rates based on volumetric soil water measures, while Yasso07 is still Yasso07 where "07" refers to the model version in question, this version was developed in 2007 and published by Tuomi et al. (2011). The difference between original ROMUL and ROMULv models was clarified on page 3 lines 23-28.*

**Major comments by Referee 1:**

1.  Although some of the these estimations might be afflicted with uncertainty (for example, a spatial extrapolation of understorey vegetation by kriging for total Finland based on only 18 plots is hardly meaningful), I think that a rough estimation of an important but difficult parameter is better than not account for it at all.

*Actually, we used much more data, and apologize it was not properly described in the earlier version of the paper. Text was modified on page 6 line 1 and on page 6 line 18 and it was made clear that **only the biomass vs biomass cover relationship was made** using data from 18 stands (a total of 504 of 0.3\*0.3 m² sample squares), while **co-kriging was based on 2501 permanent sample plots** with understorey cover measurements.*

2. There is only one important point which is not clear so far, the depth of the SOC estimates. I did not find any information about the depth for which SOC was predicted.

*Sections 2.4.2 and 2.4.3 about Yasso07 & ROMULv models were modified in order to describe how soil depth was defined with models. For Biosoil data this information was given in the section 2.5.*

3. From my point of view the structure of those models only allows a prediction for topsoils (0- 20 cm), for deeper parts stabilization mechanisms of mineral-associated SOC must be taken into account. At least, model performance should be tested not for the total depth but for different depth increments (e.g. 0-20, 20-40 cm etc.).

*We agree that models have their strength in estimation of decomposition of litter with varied quality. Both models still have compartments for stable carbon, but these carbon pools are not depth-specific. In the Yasso07 model this pool is named as "humus" box, while in the ROMULv model this is named as "stabile humus" box (see fig.1). These boxes integrate stabile carbon accumulation from various processes, including that of mineral association. Both models are used in such applications (greenhouse gas inventories, evaluation of the effects of alternative forest management practices) where estimates for deeper soil layers are needed and the models are parametrized accordingly.*

**Minor comments by Referee 1:**

Title: I would include "Forest soil carbon" and "Finland"

*The title was changed accordingly to: "Forest soil carbon stock estimates in a nationwide inventory: evaluating performance of the ROMULv and Yasso07 models in Finland"*

P1, L11: generally, the manuscript is well-written, but the first sentence of the abstract is not very good ("We test...weather data are enough..."), please rephrase.

*The beginning of the abstract was re-phrased.*

P2, L3: References for Sweden and Germany?

*References were added*

P2, L11: to reproduce

*This was corrected*

P2, L17: here is becomes clear for the first time that the paper is about forest SOC, however, this could be clarified from the beginning with a clear focus on forest SOC models and references to similar studies.

*The abstract of the paper was modified accordingly.*

P2, L20: CENTURY model

*This was corrected*

P5, L31: the kriging approach was only shortly mentioned. Due to its importance the overall performance of this important step should be described in more detail, e.g. by showing variograms.

*R code, variograms and cross-variograms of vegetation group coverages are shown in the Supplement.*

**Major comments by Referee 2:**

1. Being able to predict the current inventory is very different than being able to accurately assess changes in carbon stocks due to land use or climate. The authors more or less make this point in the introductory comments but then go on to try to model the current inventory data as a means for improving soil carbon models.

*We fully agree with this comment. Estimation of C stock change is very different than estimation of the current soil carbon stocks with these models. However, the initial soil carbon stocks should be at right level in order to predict correct changes of soil carbon in future due to fact that soil C change is relative to stock size. Therefore it is essential to have soil C stock at right level to start with simulations. We added further clarification on page 2 lines 23-25.*

2. Given the detailed inventory data already available, a spatial model seems to be the way to go to improve the national inventory. If the authors are trying to develop a model that can predict year-to-year variations in soil carbon stocks then calibrating models on a few sites with really good long-term data seems more powerful than trying to recreate mean latitudinal trends on soil carbon.

*Yes, we agree, and this is what we are aiming at, i.e. providing understanding on spatial representativeness of models by using wide spatially representative datasets. The reviewer also mentions 'year-to-year' development aspects of models. However, we're not developing a model to predict year-to-year variability of stocks, but rather evaluating models from the perspective how well they are able to replicate latitudinal patterns of soil C. The*

*performance of the models in the south – north gradients of vegetation productivity and climate is central for their wider applicability (e.g. for national GHG inventories), and implies whether the decomposition rates are at correct levels, and correctly sensitive to local mean climates. The idea of the paper is to identify locations and conditions where model performances are reduced, and if there are simple yet useful additional drivers that could be incorporated in future models based on these data. We modified objectives of the paper to be more clear, see page 3 from line 27.*

3. Additionally, most of the year-to-year change in soil carbon is going to come from land use and management decisions which have essentially been ignored here because this MS only focuses on established forests on upland soils.

*This is correct; our aim in the manuscript is to test how well these models are able to model soil carbon stocks. The ability of models to estimate C stocks at right level is a prerequisite for models to be able to model changes in soil carbon stock due to land-use changes and management events. Especially, estimates of C exchange fail on the land use conversions from forests to other uses, if original forest soil carbon stocks are systematically wrong by models. This was made clearer on page 2 lines 23-25.*

4. I really struggled with this paper. It seems that two separate ideas on model testing ending up getting merged together in this MS along with some new measurements on understory biomass and litter turnover. Given the goals of the paper, I would have thought that one model needed to be used that could have different levels of complexity added or removed. Given the fundamental differences between the Yasso07 and ROMUL models I do not see how it is possible to test the hypothesis that "accounting for soil properties" would improve model performance and then go on to say that time step (annual v. daily) might matter. The second hypothesis is only related to the ROMUL model. Since most of the hypothesis testing relates to the ROMUL model, why not just try to add a better litter module onto the ROMUL model and see if this matters?

*We agree with this point. Therefore we modified objectives of the paper and in the current version we clarify that the impact of soil texture and water holding capacity to soil carbon stock accumulation was evaluated primarily by comparing ROMULv model with and without soil water content data. We also wanted to include Yasso07 into the comparison due to fact that its decay parameters have been estimated from extensive database and mainly from Nordic countries. Comparing and seeing results of these two models in parallel is useful for GHG experts of many countries using Yasso07 for forest soils and land surface modelers (e.g. Yasso07 is implemented in JSBACH), but it lacks nitrogen dynamics rendering it incapable of simulating vegetation-soil dynamics under climate change. Text was modified on page 3 line 23-30.*

5. Model success was never really defined. The author's suggest that the ROMUL model with some information on soil water holding capacity is the superior model but based better

representation of southern soil carbon data but the goodness-of-fit statistics presented in Fig 6 are equivocal on this point.

*Text was clarified at the end of the material and methods section on page 10 line 21->. There we justify the use RMSE as a measure of model performance.*

**Minor comments by Referee 2:**

1. Abstract: need to emphasize that these models and model developments are only applicable to northern forests on upland soils. This probably goes for changing the title as well.

*Agree. The title was changed to: "Forest soil carbon stock estimates in a nationwide inventory: evaluating performance of the ROMULv and Yasso07 models in Finland". Also text in the abstract was updated.*

2. page 1. L25: "However, the significance of different drivers of soil carbon stocks is still unknown", this statement is a bit belittling to the soil carbon science community - we know a lot about what controls SOC levels.

*This is a valid point. We reformulated this sentence to be more precise and now we discuss about long term soil carbon accumulation, see page 2 line 2.*

3. page 1. L25: "On the other hand …", this sentence is not a logical juxtaposition to the previous sentence.

*Text was corrected, see from page 2 line 6.*

4. page 2, beginning: "of these inventories is usually not adequate for national-level soil carbon stock change assessment", inventories aren't meant for this purpose but many nations have or are now conducting re-sampling campaigns for this purpose.

*We agree with this comment*

5. page 2. Line 15 "The majority of countries apply soil carbon 5 models, like Yasso07 (Tuomi et al. 2011) and CENTURY (Parton et al. 1987) to estimate soil carbon stock changes. The scientific community is also aiming to predict future soil–climate change feedbacks on a global level using Earth system models (ESMs).", there is a big difference between biogeochemical models developed for plot level investigations and ESMs running at large grid scales. I'd prefer the two not be lumped into the same discussion in this paragraph. There are severe limitations in data availability globally that ESMs have to deal with.

*We agree and in the current version GHG inventory models and ESM models are introduced in separate paragraphs on page 2, lines 13-31.*

6. page 2. L10. "Individual soil carbon models are tested against repeated soil inventories and it is found that models are able to estimate soil carbon stock change of the same magnitude as was measured." , please give few citations for example

*References were added on page 2, line 27.*

7. page 2. L20. "add model between CENTURY and clay"

*This was implemented, page 3 line 3.*

8. page 3. L10. "ROMUL including the impact of nitrogen and soil water holding capacity to decomposition).", according to below you are not testing nitrogen but only water in ROMUL v. Yasso?

*This has been misleadingly reported in the manuscript. A sentence was added under 2.3 (page 7 line 18) to describe that we used constant N fractions by species and by biomass compartment. Also those nitrogen fractions were provided in the Supplement as a table.*

9. page 3. L30. "From the grid, only locations that were on upland soils and on forest, according to Food and Agriculture Organization of the United Nations (FAO) forest definitions were chosen. This classification is based on Multisource National Forest inventory products (Tomppo et al. 2008).", this is an important point and I am struggling to fully understand how this was defined (i.e., for upland soil, what soil map or was a topopgraphic property used) and the impact this has on model evealuation.

*Text was clarified and the content of the Multisource National Forest inventory product is described in the text on page 4, lines 25-26.*

10. page 4. L5. NFI9 or NFI10

*Here we used sample trees of NFI10. We had a good routine for that in hands. Results for BEFs would be the same with using sample trees from NFI9 (there could be differences in the decimals).*

11. page 4. can you give a bit more information such as goodness-of-fit statistics for this model?

*Sample size and adjusted $R^2$ were added into the text. See the end of section 2.1, page 5 line 23.*

12. page. L15. "parts, if applicable. These models were estimated separately for southern and northern Finland.", Good linear mixed models can be developed on a sample size of only 9?

*This was not properly described in the earlier version of the paper. Text was modified on page 5 and page 6 and it was made clear that biomass models were based on sampling done in total of 504 of 0.3\*0.3 m² sample squares from 18 forest stands, while co-kriging was done on 2501 permanent sample plots with understorey coverage measurements. Text was modified on page 6 line 1 and on page 6 line 18.*

13. page 7. L15. how do you get litter quality for each of the items listed in Table 2? I would think these properties would vary significantly for the same species depending on fertility status of the site.

*Text under "2.4.2 Yasso07" was modified and the reference to Yasso07 manual was added, see page 8 line 18.*

*Litter quality (solubility ratios) varied by species, but not by site types. Unfortunately we don't have litter quality data available by site types.*

14. page 7. L25 "Here, we used the Yasso07 model with Scandinavian parameters (Rantakari et al. 2012) and with global parameters, these being different from Tuomi et al., (2011) parameters.", this is awkwardly phrased. I thought you use the global parameters of Tuomi et al. but this sentence makes it sound like you did something different.

*Text under "2.4.2 Yasso07" was modified and we explain that our parameter set is a preliminary version from those published by Tuomi et al. 2011. This set provides practically same results as those published and is used here due to fact that it has been earlier used by the Finnish GHG inventory. With MCMC parameter estimation methods one never gets exactly same results with models that have several parameters. See page 6 line 26.*

15. page 8. L5. "We applied the ROMUL model using daily time-steps of the environmental variables impacting the decomposition.", how were litter inputs imputed at a daily timestep

*Litter was distributed evenly for each day of the year. This procedure was chosen due to unknown timing of the belowground litter and due to fact that we are here interested about long term C accumulation, not short term fluctuations of C between forests and atmosphere. Text was added on page 9 lines 19-20.*

16. page 8. L20. What depth are the models assumed to be working to? This will really impact how much root inputs there will be and the SWHC data including the 20/80 split of SWHC between O and M soil.

*Sections 2.4.2 and 2.4.3 about Yasso07 & ROMULv models were modified in order to describe how soil depth was defined with models. For Biosoil data this information was given in the section 2.4.4. See page 8 line 28 and page 9 23-25.*

17. page 9. L15. there is a lot of discussion material in the results section. I do not mind combined R&D sections but it has to be one or the other.

*Results section was re-organized. We introduced subtitles there and we removed texts that were discussing the findings with understorey litter. See pages 10 - 12.*

18. page 9. L15.  given this MS is focused on soil carbon model development not biomass model development, perhaps consider moving Fig 2 and 3 and Table 3 to supplemental material.

*We agree with this comment, those graphs and that table were moved into the Supplement of the manuscript.*

19. page 9. L25.  "Our estimate of mean litter input from trees was 1962 kg ha-1 and 903 kg ha-1 of carbon for southern and northern Finland, respectively", are these values reasonable when compared to estimates of NPP?

*Comparison to modeled NPP values and measured litter fall data were added into the discussion under section 4, page 12 from line 32.*

20. page 10. L5. "the southernmost plots (Fig 3).", wrong figure reference"

*This was corrected, page 11 line 27.*

21. page 10. L30. "We also found that litter input from understorey vegetation is equal of that from trees in northern Finland. This", this seems like an important finding and shouldn't be buried within a model development paper.

*We re-organized results section and currently we have there two subtitles, 3.1. Performance of the soil carbon models and 3.2. Improved understorey litter input. We hope that using subtitles brings up the finding of the high understorey litter input in northern Finland. See page 11 lines 10-17 and page 12 line 13.*

22. page 11. L1. "To sum up, litter input from understorey vegetation has to be quantified properly when soil carbon models are parameterised.", but this only mattered when the local parameterization of the model was used. Including understory litter actually decreased the performance of the global model.

*The reason why the performance of the global model was reduced with understorey litter is the fact that parametrization of the model has lacked that larger input of understorey litter (found in this paper) into the soil system. And now providing correct litter input (which is more than earlier thought) results higher C stocks compared to those measured. This is our point here. And we added a sentence on page 13 line 4-5 to clarify this.*

23. page 11. L20. "This result indicated that the quantity of precipitation alone was not a sufficient modifier for decomposition but when complemented with soil water holding capacity results improved. This finding supports the use of models including soil texture and

water holding capacity.", latitudinally averaged results improved but the site to site variability seems to have increased dramatically.

*This is true and we added "average latitudinal results" into the text, see page 13 line 25. Site to site variability increases due to fact that variation in the soil properties cause very different decomposition conditions under varying precipitation schemes (see Fig. 4).*

24. page 11. L20. "According to our results, the time step of the simulation plays a critical role; running", BUT there were so many other differences between models that this is a hard conclusion to reach. You would have to run the same model at the different time steps to assess this.

*We agree, text was modified and now it says that our results support earlier findings, where …. This modification was done on page 13 from line 27.*

25 page 12. L1. "The use of water holding capacity was critical for accurate soil carbon stock estimation by the ROMUL model…", you never defined what success looks like. An R2 of 0.368 is considered success but an R2 of 0.367 is not? Or are you using RMSE? I would argue slope is perhaps the most important.

*We added text in the end of material and methods section, where we say: "We used root mean square error (RMSE) to rank these models applications due to its ability to take into account both accuracy and precision when comparing model estimates and data." See page 10 from line 21.*

*Thereafter all judgements about performance of the models have been based on that.*

26. page 12. L5. "(Fig. 6). The fact that Yasso07 soil carbon stocks were more accurate when litter input of understory vegetation was omitted from simulations suggests that the model calibration did not account for the whole range of understory litter input.", OR there are other important drivers (such as soil properties) that are missing from the model.

*We mean here that if Yasso07 would have been originally parametrized with our understorey litter input, thereafter it would have been able to model latitudinal trends of soil carbon in Finland. We added a sentence after, where we explain the meaning of this finding. See page 14 line 11.*

27. page 12. L5 "However, three model simulations out of six produced relatively accurate", how defined? this is not a very quantitative term.

*This is true. We modified the text on page 14 line 16. And we added a RMSE threshold there.*

[revised manuscript text omitted]

A. Lehtonen et al.

Correspondence: A. Lehtonen (aleksi.lehtonen@luke.fi)

**1. Fineroot turnoverate modeling**

Fineroot turnoverrates were based on Kleja et al. (2008) and to Leppälammi-Kujansuu et al. (2014). It was assumed that the turnoverrate of fineroots is 85% in Southern Finland in areas where temperature sum during the growing season (with 5 Cº treshold) is more than 1200. For Northern Finland we used turnoverate of 50% when temperature sum was less than 700. For sites where temperature sums were between 700 and 1200 we interpolated the turnoverrate with a simple linear regression,

$y = b_0 + b_1 \times x$, where

$y$ is the turnoverrate and $b_0$ equals 0.0007, $b_1$ equals 0.001 and $x$ is degree days that varies between 700 and 1200 degrees.

**2. Estimation of understorey biomass**

Here we provide parameter estimates for understorey models (Table S1) and figures for model fits (Figs. S1 & S2).

Table S1. Parameter estimates for understorey models by species groups and regions. Where $\beta_0$ is the intercept and $\beta_1$ is a slope of the fixed part of the model and bc is the bias correction. Var $b_0$, Cov $b_0 b_1$ and Var $b_1$ originate from variance-covariance matrix of the fixed part of the model. Var pop lists unexplained variance and Var plt lists unexplained variance after plot as a random effect.

| Group | Com. | Reg. | $\beta_0$ | $\beta_1$ | bc | Var$\beta_0$ | Cov$\beta_0 \beta_1$ | Var$\beta_1$ | Var Pop | Var Plt |
|---|---|---|---|---|---|---|---|---|---|---|
| Dwarf shrub | Ab | NF | 3.62 | 0.948 | 0.135 | 0.115 | -0.028 | 0.008 | 0.285 | 0.136 |
| Dwarf shrub | Ab | SF | 2.653 | 1.107 | 0.186 | 0.081 | -0.015 | 0.005 | 0.45 | 0.16 |
| Grass | Ab | - | 2.75 | 0.918 | 0.292 | 0.129 | -0.03 | 0.01 | 0.688 | 0.273 |
| Herb | Ab | - | 1.116 | 1.08 | 0.445 | 0.074 | -0.018 | 0.008 | 0.832 | 0.572 |
| Bryophytes | Ab | - | 3.13 | 0.795 | 0.281 | 0.301 | -0.072 | 0.018 | 0.633 | 0.306 |
| Lichen | Ab | - | 3.69 | 0.894 | 0.109 | 0.047 | -0.012 | 0.005 | 0.27 | 0.175 |
| Shrub | Bl | NF | 6.278 | 0.45 | 0.161 | 0.086 | -0.02 | 0.006 | 0.315 | 0.241 |
| Shrub | Bl | SF | 3.73 | 0.831 | 0.458 | 0.15 | -0.035 | 0.011 | 1.06 | 0.867 |
| Grass | Bl | NF | 3.368 | 0.589 | 0.639 | 0.169 | -0.039 | 0.017 | 1.423 | 0.859 |
| Herb | Bl | SF | 1.877 | 0.906 | 0.709 | 0.27 | -0.067 | 0.03 | 1.195 | 0.898 |

Com. = compartment: Ab = aboveground, Bl = belowground
Reg. = region: NF = northern Finland, SF = southern Finland

[Figure]

**Figure S1: Models and data points for aboveground understorey biomass (A = Dwarf shrubs in northern Finland, B = Dwarf shrubs in southern Finland, C = Grasses, D = Herbs, E = Mosses and F = Lichen). Solid line based on modelling that takes into account site fertility distribution for Finland by weighting, while dashed line is the estimate without weights.**

[Figure]

**Figure S2. Models and data points for belowground understorey biomass (A = Dwarf shrubs in northern Finland, B = Dwarf shrubs in southern Finland, C = Grasses and D = Herbs). Solid line based on modelling that takes into account site fertility distribution for Finland by weighting, while dashed line is the estimate without weights.**

R code for variograms and cross-variograms of vegetation group coverages and resulting figure with semivariance as a function of distance (Fig. S3).

```
library(gstat)

library(sp)

**peite34 object includes data for vegetation coverages and plot locations across Finland**

coordinates(peite34) = c('x','y')

####################

g.r <- gstat(NULL, "dwarf shrubs", r1~x+y,data = peite34,  model = v.fitr1, nmax=200)

g.r <- gstat(g.r, "grass&herb", r2~x+y,data = peite34,  model = v.fitr2, nmax=200)

g.r <- gstat(g.r, "bryophyte", r3~x+y,data = peite34,  model = v.fitr3, nmax=200)

g.r <- gstat(g.r, "lichen", r4~x,data = peite34,  model = v.fitr4, nmax=200)

vm <- variogram(g.r)

vm.fit <- fit.lmc(vm,g.r,vgm(1200,"Sph",100,300))

plot(vm, vm.fit)

**thereafter predictions of the understorey coverage for each location by vegetation groups**

cok.maps <- predict(vm.fit, nfi10loc)
```

[Figure]

**Figure S3. Variograms and cross-variograms for different functional types of understorey vegetation.**

**3. Nitrogen content of the litterfall**

Table S2. Nitrogen content of litterfall as ratios from mass, based on Komarov et al. (2007).

|  | Foliage | Branches | Stem wood | Fine roots | Coarse roots | Total |
|---|---|---|---|---|---|---|
| Scots pine | 0.003 | 0.004 | 0.0014 | 0.0047 | 0.0024 | - |
| Norway spruce | 0.0045 | 0.0035 | 0.002 | 0.0035 | 0.003 | - |
| Deciduous | 0.007 | 0.004 | 0.0015 | 0.005 | 0.0045 | - |
| Understorey[1] | - | - | - | - | - | 0.006 |

**4. Soil carbon model Yasso07**

Parameters for the Yasso07 have been estimated with Markov Chain Monte Carlo (MCMC) methods where litter bag, wood decomposition and soil carbon stock data were used to calibrate model parameters. Maximum posterior parameters estimates used in this study were based on works by Rantakari et al. (2012) and for earlier version of the global Yasso07 parameterisation that is a close
* * *
[1] The nitrogen content of the understorey litterfall was estimated to be 0.6% from dry mass.

variant of the Tuomi et al. (2011) publication (Table S2).  Model have been described more in detail in papers where individual components have been reported (Tuomi et al. 2009, Tuomi et al. 2011, Tuomi et al. 2008)

Fortran code of the Yasso07 model is available here:

http://code.google.com/p/yasso07ui/

Table S2. Yasso07 maximum *a posterior* (MAP) point estimates Scandinavian (Rantakari et al. 2012) and global parameterization (Tuomi et al. 2011).

| Parameter | Scandinavia | Global | Unit | Meaning |
|:---:|:---:|:---:|:---:|:---:|
| aA | 0.52 | 0.72 | $a^{-1}$ | decomposition rate of A |
| aW | 3.55 | 5.9 | $a^{-1}$ | decomposition rate of W |
| aE | 0.35 | 0.28 | $a^{-1}$ | decomposition rate of E |
| aN | 0.27 | 0.031 | $a^{-1}$ | decomposition rate of N |
| p1 | 0.04 | 0.48 | . | mass flow from W to A |
| p2 | 0.03 | 0.01 | . | mass flow from E to A |
| p3 | 0.98 | 0.83 | . | mass flow from N to A |
| p4 | 0.64 | 0.99 | . | mass flow from A to W |
| p5 | 0.31 | 0.00 | . | mass flow from E to W |
| p6 | 0.019 | 0.01 | . | mass flow from N to W |
| p7 | 0.023 | 0.00 | . | mass flow from A to E |
| p8 | 0.01 | 0.00 | . | mass flow from W to E |
| p9 | 0.001 | 0.02 | . | mass flow from N to E |
| p10 | 0.34 | 0.00 | . | mass flow from A to N |
| p11 | 0.042 | 0.015 | . | mass flow from W to N |
| p12 | 0.09 | 0.95 | . | mass flow from E to N |
| b1 | 0.09 | 0.95 | $C^{-1}$ | temperature dependence parameter |
| b2 | -0.0023 | -1.4 | $10^{-}$ | temperature dependence parameter |
| y | -2.94 | -1.21 | $m^{-1}$ | precipitation dependence parameter |
| pH | 0.15 | 4.5 | $10^{-3}$ | mass flow from A,W,E,N to humus |
| aH | -0.24 | -1.6 | $10^{-3}$ | humus decomposition coefficient |
| roo1 | -0.539 | -1.71 | $cm^{-1}$ | size dependence parameter |
| roo2 | 1.186 | 0.86 | $cm^{-2}$ | size dependence parameter |
| r | -0.264 | -0.306 | . | size dependence parameter |

**5. *Soil carbon model ROMULv**

ROMUL decomposition model describes the flux of soil organic matter (SOM) through the soil

decomposition process, divided into separate parallel paths of matter based on the different origins of the litter (Chertov and Komarov 1997, Chertov et al. 2001). All fluxes have essentially the same pattern of flux. The litter entering the decomposition process is stored in a store of undecomposed litter (H). This then decomposes into a mixture of partly decomposed (humified) SOM (F). The SOM in the F fraction is decomposed by different types of organisms (fungi, bacteria and earthworms) and end up in storage of semi-stable humus (H). These fluxes (for litter of different origins) are for SOM, but the model also describes a parallel decomposition process for nitrogen in soil, following the same pattern of storages and fluxes.

Litter $\rightarrow$ L $\rightarrow$ F $\rightarrow$ H

In each transition from one storage to another, the rate of decomposition depends on the nitrogen content of the storage (calculated from the ratio of the SOM storage and the corresponding N storage) and the ash content of the litter (fixed percentage for each litter source). These transfer rates are adjusted with coefficient functions depending on soil temperature and soil water content. During each transition, a fraction of the pool is mineralized – SOM released as $CO_2$ and N released into a pool of mineralized N available to plants, while most of the matter moves to the next pool in the decomposition process. The litter from different sources follow different routes, until they all end up in a common pool of semi-stable humus (H). Finally, also the matter in the H pool slowly decomposes. In each transition from one pool to another, the rate of SOM and N decomposition is the same, except for the transition from F pool to H; here the C:N ratio of the transition is characteristic for all decomposer classes (fungi, bacteria and earthworms), and in this phase the N content of the soil is enriched.

The litter from aboveground (leaves, shoots and trunks) fall onto the forest floor, whereas the root litter (fine roots and coarse roots) decomposes in the mineral soil layer. For the two layers, the decomposition rates based on SOM properties are mostly similar, but the environmental conditions (temperature and soil moisture) may differ for the two layers, resulting in different decomposition rates. The semi-stable storage H is common for all fluxes and resides in the mineral soil layer.

The original ROMUL model uses decomposition rate functions that use gravimetric measures of soil

water as input. As these figures are somewhat tedious to calculate and would require detailed information of the soil characteristics, such as soil density and precise thickness of the soil layers, we used instead a modification presented by Linkosalo et al. (2013) named here as ROMULv. In their paper they produced decomposition rate functions that are based on volumetric soil water measures, and a volumetric soil water model to predict the variation in the soil water content, driven by environmental measures. The volumetric soil water model was easier to apply for multiple simulation points, as it only needs the total potential soil water storage (i.e. difference between field capacity and wilting point) to characterize the soil for the simulations of relative soil water content.

The description of the original ROMUL model is available from here:

http://ecomodelling.ru/

For the source code used in this study, see the end of this document.

**Source code of the ROMULv model used here**

```
Main program loop (meta-code):

Initialize ROMUL (read initial values from file)

Loop_over_data {

  read meteorological data
  calculate soil temperature

  calculate evapotranspiration
  calculate soil water status

  calculate decomposition and update soil storages
}

write ROMUL results
```

```
cccccccccccccccccccccccccccccccccccccccccccccccccccccccccccccccccccc
c
c      This implementation of the ROMUL model is based on
c      Chertov et al. 2001, with "new functions" from
c      Komarov et at. 2008.
c
c      Tapio Linkosalo (METLA) & Annikki Mäkelä (UH)
c
cccccccccccccccccccccccccccccccccccccccccccccccccccccccccccccccccccc
```

```
          subroutine ROMUL(litter,litter_N,T,SW,Navail,
     +                site, year, init)

          implicit none

c litterfall by cohort, soil temperature and water content for organic (==1)  and mineral soil layer
(==2)
c init-parameter for initializing the model (1), normal simulation (0) or output the results (2).

          double precision litter(11), litter_N(11)
          real T(2), SW(2)

c mass storages are save-variables, values persist from one call to another
c First dimension of pools is for the SOM cohort, the second for spatial locations
c (except only one pool per geographic location for humus)
          double precision Lpool(11,4000), Fpool(11,4000)
          double precision LNpool(11,4000), FNpool(11,4000)
          double precision Hpool(4000), HNpool(4000)
          save Lpool, Fpool, Hpool, LNpool, FNpool, HNpool

          double precision Navail(4000)
           real time, Nup, fnf, gnf, fnwb, Ndemand, qctot
           integer site, year, init

          integer local_year
          static local_year

c declare decomposition coefficient functions
          real f_1, f_2, f_3, f_4, f_5, f_6
          real g_1, g_2, g_3, g_4, g_5, g_6
c
c * N pools etc
c
c  litter in 11 separate pools
c  all have L and F fractions, one H fraction
c  all fractions have SOM and N
c  cohorts are order thus:
c    1   foliage
c    2   branches
c    3   stems
c    4   fine roots
c    5   coarse roots
c    11  ground vegetation
c  Note! Cohorts 6 to 10 are for felling residues, but in this version
c  they are calculated in corresponding litter cohorts (6->1 etc.)

c parameters - following ROMUL
c
c      kL              specific rate of leaching (yr-1)     1.0
```

```
c       kDL(i)  specific rate of litter decomposition (yr-1)        0.3

c       kDF(i)  specific rate of SOM decomposition (yr-1)     0.3

c       kTL(i)  specific rate of transfer from litter to SOM  (yr-1)

c       kTF(i)  specific rate of transfer from SOM to humus, comp. 1  (yr-1)        0.3

c       kTF2(i) specific rate of transfer from SOM to humus, comp. 2  (yr-1)        0.3

c       kDH             specific rate of humus decomposition (yr-1) 1/6000

c       kUG             maximum specific N uptake rate of ground vegetation (yr-1) 100

c       DeposN  N deposition (free input) (kg yr-1) 1

c

c    ash_cont(i) ash content of cohorts

c    fnf     foliar N concentration in live foliage

c     gnf           retention of foliar N when shedding foliage 0.45

c

c

cccccccccccccccccccccccccccccccccccccccccccccccccccccccccccccccc

c Declare internal variables

      double precision DeposN, LeachN
       double precision d_Lpool(11),d_Fpool(11),d_HumusPool
      double precision d_LNpool(11),d_FNpool(11),d_HumusNPool
       double precision d_Navail
      double precision fn(11)
       real kL, kM, DM
      double precision kDL(11), kDH
      double precision kDF(11)
      double precision kTL(11)
      double precision kTF(11),kTF2(11)
      real LKP_decomp
      double precision K_1S_rma, K_2S_rma,K_3S_rma, k_1L_rma,K_2L_rma
      double precision k_3L_rma, k_4_rma,k_5_rma,k_6_rma
      double precision ML, MF(11), MFres(11), MH, gamma, GVN1, GVN2
       double precision N_release, FH_FluxB, FH_fluxL, FHN_FluxB, FHN_fluxL
      double precision MF_flux, MF_min
      double precision H_miner, H_C, C_C, DeltaB, DeltaL, dL
      double precision x1, x2, ash_cont(11)
      integer inttime, i, j, k, inttimetot
      double precision Lpoolsum(5),Fpoolsum(5),LNpoolsum(5),FNpoolsum(5)
      double precision littersum(5), Fpooltot(2), FNpooltot(2), H_CN
      double precision Lpooltot(2), LNpooltot(2)
      double precision NFconc(2), NLconc(2)
       double precision step

c assign parameter values
       data kL
     1 /0.12/

      data (ash_cont(i) , i=1,11)
     1 /0.02,  0.02,  0.01,  0.02,  0.02, 0.02, 0.02, 0.01,
```

```
      1 0.02, 0.01, 0.04/

       data fnf, gnf  /0.02, 0.45/

       data DeposN, kM, DM /2., 2., 0.3/
       data ML, gamma, C_C / 0.1, 0.8, 0.5/
       data deltaB, deltaL /24., 12.8/

       dL = 1

c read inputs when coming to this subroutine for the first time (init = 1)
      if (init.eq.1) then
          open(unit=25,file='soil.dat',status = 'old', err = 999)
         do 100 i = 1, site
          read(25,*) j, (Lpool(j,i),j=1,5),Lpool(11,i),
     +                      (Fpool(j,i),j=1,5),Fpool(11,i),
     +                         Hpool(i),
     +             (LNpool(j,i),j=1,5),LNpool(11,i),
     +             (FNpool(j,i),j=1,5),FNpool(11,i),
     +             HNpool(i)

100      continue
          close(25)
         goto 998
999      write(*,*) "failed to open soil.dat"

          local_year = 0

998         return
      endif

c write storages to file when coming to this subroutine for the last time (init = 2)
      if (init.eq.2) then
          open(unit=25,file='soil.dat',status = 'old')

         write(*,*) "writing soil.dat..."
         do 101 i = 1, site
             write(25,997) i,(Lpool(j,i),j=1,5),Lpool(11,i),
     +                       (Fpool(j,i),j=1,5),Fpool(11,i),
     +            Hpool(i),
     +          (LNpool(j,i),j=1,5),LNpool(11,i),
     +          (FNpool(j,i),j=1,5),FNpool(11,i),
     +              HNpool(i)
997  format(i5,12f12.2,f14.2,12f12.5,f14.3)
```

```
101      continue
          close(25)
         return
       endif

c Daily calculation starts here!

         do 65 i = 1, 11
             fn(i) = litter_N(i)/litter(i)
65       continue

c "step" is for substeN of differential calculation, with daily weather data
c no substeN needed.
           step = 1

c compute specific rate parameters as functions of ash content and N content
c compute total N in litter

c first the N concentration for the F-pools:
c index 1 = organic layer, and 2 = mineral soil

      Lpooltot(1) = Lpool(1,site) + Lpool(2,site) + Lpool(3,site) +
     +               Lpool(11,site)
      LNpooltot(1) = LNpool(1,site) + LNpool(2,site) + LNpool(3,site) +
     +               LNpool(11,site)
      Fpooltot(1) = Fpool(1,site) + Fpool(2,site) + Fpool(3,site) +
     +               Fpool(11,site)
      FNpooltot(1) = FNpool(1,site) + FNpool(2,site) + FNpool(3,site) +
     +               FNpool(11,site)
      Fpooltot(2) = Fpool(4,site) + Fpool(5,site)
      FNpooltot(2) = FNpool(4,site) + FNpool(5,site)

      NFconc(1) = FNpooltot(1) / Fpooltot(1)
      NFconc(2) = FNpooltot(2) / Fpooltot(2)
      NLconc(1) = LNpooltot(1) / Lpooltot(1)
      NLconc(2) = LNpooltot(2) / Lpooltot(2)

      do 25 i = 1,5

          if(i.ge.4) then
             kDL(i) = k_1S_rma(ash_cont(i), fn(i)) *
     *            f_1(T(2)) * LKP_decomp(SW(2), 0.55)
            kDF(i) = k_2S_rma(ash_cont(i), NFconc(2)) *
     *            f_2(T(2)) * LKP_decomp(SW(2), 0.55)
             kTL(i) = k_3S_rma(ash_cont(i), fn(i)) *
     *            f_3(T(2)) *LKP_decomp(SW(2), 0.55)
          else
```

```fortran
            kDL(i) = k_1L_rma(ash_cont(i), fn(i)) *
     *           f_1(T(1)) * LKP_decomp(SW(1), 0.7)
            kDF(i) = k_2L_rma(ash_cont(i), NFconc(1)) *
     *           f_2(T(1)) * LKP_decomp(SW(1), 0.7)
            kTL(i) = k_3L_rma(ash_cont(i), fn(i)) *
     *           f_3(T(1)) * LKP_decomp(SW(1), 0.7)
          endif

        kTF(i) = k_4_rma(ash_cont(i), NFconc(2)) *
     *           f_4(T(2)) * LKP_decomp(SW(2), 0.55)
        kTF2(i) = k_5_rma(ash_cont(i), NFconc(2)) *
     *           f_5(T(2)) * LKP_decomp(SW(2), 0.55)

 25     continue

        kDL(11) = k_1L_rma(ash_cont(11), fn(11)) *
     *           f_1(T(1)) * LKP_decomp(SW(1), 0.7)
        kDF(11) = k_2L_rma(ash_cont(11), NFconc(1)) *
     *           f_2(T(1)) * LKP_decomp(SW(1), 0.7)
        kTL(11) = k_3L_rma(ash_cont(11), fn(11)) *
     *           f_3(T(1)) * LKP_decomp(SW(1), 0.7)

        kTF(11) = k_4_rma(ash_cont(11), NFconc(2)) *
     *           f_4(T(1)) * LKP_decomp(SW(1), 0.55)
        kTF2(11) = k_5_rma(ash_cont(11), NFconc(1)) *
     *           f_5(T(1)) * LKP_decomp(SW(1), 0.55)

       kDH = k_6_rma() * f_6(T(2)) * LKP_decomp(SW(2), 0.55)

c       update N pools

        Nup = 0.

c     do 100 inttime = 1, inttimetot

      LeachN = kL * Navail(site)
      LeachN = max(LeachN, 0.)

c C pools: derivatives

      FH_fluxB = 0.
      FH_fluxL = 0.
      FHN_fluxB = 0.
```

```fortran
      FHN_fluxL = 0.

       do 10 i = 1,5

           d_Lpool(i) = litter(i) + litter(i+5) -
     -          (kDL(i) + kTL(i)) * Lpool(i,site)

           d_Fpool(i) = kTL(i) * Lpool(i,site)
     1                     -(kDF(i) + kTF(i) + kTF2(i)) * Fpool(i,site)

           FHN_fluxB = FHN_fluxB + kTF(i) * FNpool(i,site)
           FHN_fluxL = FHN_fluxL + kTF2(i) * FNpool(i,site)
           FH_fluxB = FH_fluxB + kTF(i) * Fpool(i,site)
           FH_fluxL = FH_fluxL + kTF2(i) * Fpool(i,site)

 10      continue

           d_Lpool(11) = litter(11)  -
     -          (kDL(11) + kTL(11)) * Lpool(11,site)

           d_Fpool(11) = kTL(11) * Lpool(11,site)
     1                     -(kDF(11) + kTF(11) + kTF2(11)) * Fpool(11,site)

           FHN_fluxB = FHN_fluxB + kTF(11) * FNpool(11,site)
           FHN_fluxL = FHN_fluxL + kTF2(11) * FNpool(11,site)
           FH_fluxB = FH_fluxB + kTF(11) * Fpool(11,site)
           FH_fluxL = FH_fluxL + kTF2(11) * Fpool(11,site)

            d_HumusPool = DeltaB * FHN_fluxB + DeltaL * FHN_fluxL
     +                 - kDH * Hpool(site)

c calculate MF(i)

       do 11 i = 1,5

           if((100.*NFconc(2) - 1.16 * 100.*NFconc(1)) . le. 0.44) then
               MF(i) = 0.1
           else
               if((100.*NFconc(2) - 1.16*100.*NFconc(1)) . le. 1.50) then
                   MF(i) = 0.5
               else
                   MF(i) = 1.0
               endif
           endif
```

```
11    continue

          if((100.*NFconc(2) - 1.16 * 100.*NFconc(1)) . le. 0.44) then
              MF(11) = 0.1
          else
              if((100.*NFconc(2) - 1.16*100.*NFconc(1)) . le. 1.50) then
                  MF(11) = 0.5
              else
                  MF(11) = 1.0
              endif
          endif

c MF_flux is transferred from F to H, MF_min is released

      MF_flux = 0.
      MF_min = 0.

      do 20 i = 1,5

          d_LNpool(i) = litter_N(i) + litter_N(i+5)
     1                    - (ML*kDL(i) + kTL(i)) * LNpool(i,site)

          d_FNpool(i) = (kTL(i)) * LNpool(i,site)
     1                    - MF(i) * kDF(i) * FNpool(i,site)
     1                    -(kTF(i) + kTF2(i)) * FNpool(i,site)

          MF_flux = MF_flux + (1-MF(i)) * kDF(i) * FNpool(i,site)
          MF_min = MF_min + MF(i) * kDF(i) * FNpool(i,site)

          Lflux(i,site) = Lflux(i,site) + ML*kDL(i) * LNpool(i,site)
          Fflux(i,site) = Fflux(i,site) + MF(i)*kDF(i) * FNpool(i,site)

20    continue

          d_LNpool(11) = fn(11) * litter(11)
     1                    - (ML*kDL(11) + kTL(11)) * LNpool(11,site)

          d_FNpool(11) = (kTL(11)) * LNpool(11,site)
     1                    - MF(11) * kDF(11) * FNpool(11,site)
     1                    -(kTF(11) + kTF2(11)) * FNpool(11,site)
```

```
          MF_flux = MF_flux + (1-MF(11)) * kDF(11) * FNpool(11,site)
          MF_min = MF_min + MF(11) * kDF(11) * FNpool(11,site)

       if(HNpool(site) .gt. 0.) then
          H_CN = Hpool(site) / HNPool(site) / 2.
       else
          H_CN = 100.
       endif

       if(H_CN .gt. 8.) then
          MH = 0.8
       else
          MH = 1.
       endif

          d_HumusNPool =
      +                   - kDH * MH * HNpool(site)
      1                   + gamma * (FHN_fluxB + FHN_fluxL)

C  Calculate mineralised carbon

       H_miner = kDH * Hpool(site) + FH_fluxB + FH_fluxL
      1                   - DeltaB * FHN_fluxB - DeltaL * FHN_fluxL
      1                   - DeltaB * kM * Navail(site)

       do 15 i = 1,5
          H_miner = H_miner + kDL(i) * Lpool(i,site)
      +               + kDF(i) * Fpool(i,site)
15     continue

          H_miner = H_miner + kDL(11) * Lpool(11,site)
      +               + kDF(11) * Fpool(11,site)

       H_C = C_C * H_miner
       C_sum(site) = C_sum(site) + H_C

c calculate N release

       N_release =    kDH * MH * HNpool(site)
      1               + (1. - gamma) *(FHN_fluxB + FHN_fluxL)
      1                 + DeposN
c     1                  - kM * Navail(site) + DeposN
```

```fortran
      Hflux(site) = Hflux(site) + kDH* MH * HNpool(site)

      do 16 i = 1,5
          N_release = N_release + ML * kDL(i) * LNpool(i,site)
     1                + MF(i) * kDF(i) * FNpool(i,site)

16    continue

          N_release = N_release + ML * kDL(11) * LNpool(11,site)
     1                + MF(11) * kDF(11) * FNpool(11,site)

       d_Navail = N_release  - LeachN
c     1              - N_coeff_TR * Ndemand
c     1              - N_coeff_GV1 * Gr_GVegN1 - N_coeff_GV2 *  Gr_GVegN2

c Update pools - use simple Euler

      do 17 i = 1, 5
c daily version -> step equals one!
          Lpool(i,site) = Lpool(i,site) + d_Lpool(i) * step
          Fpool(i,site) = Fpool(i,site) + d_Fpool(i) * step
          LNpool(i,site) = LNpool(i,site) + d_LNpool(i) * step
          FNpool(i,site) = FNpool(i,site) + d_FNpool(i) * step
17    continue

          Lpool(11,site) = Lpool(11,site) + d_Lpool(11) * step
          Fpool(11,site) = Fpool(11,site) + d_Fpool(11) * step
          LNpool(11,site) = LNpool(11,site) + d_LNpool(11) * step
          FNpool(11,site) = FNpool(11,site) + d_FNpool(11) * step

      HPool(site) = HPool(site) + d_HumusPool * step
       HNPool(site) = HNPool(site) + d_HumusNPool * step
       Navail(site) = Navail(site) + d_Navail * step

      N_sum(site) = N_sum(site) + d_Navail * step

c      Navail(site) = max(Navail(site), 0.)

      return
      end subroutine ROMUL
```

```
c       --------------------------------------------------------------------------------
c       The functions for rate of decomposition modifiers depending on temperature
c       of the corresponding cohort. These new functions are from Komarov et al. 2008.
c
c       Tapio Linkosalo October 2008
c       --------------------------------------------------------------------------------
        real function f_1(T)

                real T

                if (T .LE. -5.0 .OR. T .GT. 60.0) then
                        f_1 = 0
                endif
                if (T .GT. -5.0 .AND. T .LE. 1.0) then
                        f_1 = 0.1595 + 0.0319 * T
                endif
                if (T . GT. 1.0 .AND. T .LE. 35.0) then
                        f_1 = 0.1754 * exp(0.0871 * T)
                endif
                if (T . GT. 35.0 .AND. T .LE. 60.0) then
                        f_1 = 8.791 - 0.1465 * T
                endif

                return
        end
c       --------------------------------------------------------------------------------
        real function f_2(T)

                real T

                if (T .LE. -5.0 .OR. T .GT. 60.0) then
                        f_2 = 0
                endif
                if (T .GT. -5.0 .AND. T .LE. 1.0) then
                        f_2 = 0.1595 + 0.0319 * T
                endif
                if (T . GT. 1.0 .AND. T .LE. 35.0) then
                        f_2 = 0.1754 * exp(0.0871 * T)
                endif
                if (T . GT. 35.0 .AND. T .LE. 60.0) then
                        f_2 = 3.690 - 0.0615 * T
                endif

                return
        end
c       --------------------------------------------------------------------------------
        real function f_3(T)
```

```fortran
            real T

            if (T .LE. -3.0) then
                  f_3 = 0
            endif
            if (T .GT. -3.0 .AND. T .LE. 7.0) then
                  f_3 = 1.3
            endif
            if (T .GT. 7.0 .AND. T .LE. 60.0) then
                  f_3 = 1.472 - T * 0.0245
            endif
            if (T. GT. 60.0) then
                  f_3 = 0
            endif

            return
      end
c     ------------------------------------------------------------------------------
      real function f_4(T)

            real T

            if (T .LE. -5.0) then
                  f_4 = 0
            endif
            if (T .GT. -5.0 .AND. T .LE. 1.0) then
                  f_4 = 0.1595 + 0.0319 * T
            endif
            if (T .GT. 1.0 .AND. T .LE. 20.0) then
                  f_4 = 0.1754 * exp(0.0871 * T)
            endif
            if (T .GT. 20.0 .AND. T .LE. 40.0) then
                  f_4 = 1
            endif
            if (T .GT. 40.0 .AND. T .LE. 80.0) then
                  f_4 = 2.0 - 0.025 * T
            endif
            if (T. GT. 80.0) then
                  f_4 = 0
            endif

            return
      end
c     ------------------------------------------------------------------------------
      real function f_5(T)

            real T

            if (T .LE. -5.0) then
```

```fortran
                        f_5 = 0
                endif
                if (T .GT. -5.0 .AND. T .LE. 1.0) then
                        f_5 = 0.078 + 0.0156 * T
                endif
                if (T .GT. 1.0 .AND. T .LE. 13.0) then
                        f_5 = 0.0675 * exp(0.2088 * T)
                endif
                if (T .GT. 13.0 .AND. T .LE. 25.0) then
                        f_5 = 1
                endif
                if (T .GT. 25.0 .AND. T .LE. 50.0) then
                        f_5 = 2.0 - 0.04 * T
                endif
                if (T. GT. 50.0) then
                        f_5 = 0
                endif

                return
        end
c       --------------------------------------------------------------------------------
        real function f_6(T)

                real T

                if (T .LE. -5.0) then
                        f_6 = 0
                endif
                if (T .GT. -5.0 .AND. T .LE. 1.0) then
                        f_6 = 0.1595 + 0.0319 * T
                endif
                if (T .GT. 1.0 .AND. T .LE. 27.5) then
                        f_6 = 0.1754 * exp(0.0871 * T)
                endif
                if (T .GT. 27.5 .AND. T .LE. 35.0) then
                        f_6 = 1.95
                endif
                if (T .GT. 35.0 .AND. T .LE. 60.0) then
                        f_6 = 4.68 - 0.078 * T
                endif
                if (T. GT. 60.0) then
                        f_6 = 0
                endif

                return
        end
c       --------------------------------------------------------------------------------
c
c       Decomposition rate function for volumetric soil water content (theta) (where
c       theta = 0 == wilting point and theta = 1 == saturation), based on paper
```

```fortran
c     Linkosalo, Kolari & Pumpanen 2013.
c
c     --------------------------------------------------------------------------------
      real function LKP_decomp(theta, porosity)

          real  theta, porosity, P1, P2

          P1 = 3.83 * theta ** 1.25
          P2 = 4.43 * (1-theta)**0.8854
          LKP_decomp = min(P1,P2,1.)

          return
          end

c     --------------------------------------------------------------------------------
c     The following are the k coefficients for the decomposition rate, depending on
c     litter ash and N content. The two values are given as parameters (absolute
c     values g/g, NOT percentage as in Romul equations!) so that the same functions
c     can be used whether the parameter values are for a specific cohort or litter
c     in general. These are the "new" functions as in Komarov et al. 2008.
c
c     TL October 2008.
c     --------------------------------------------------------------------------------
      double precision function k_1L_rma(ash, N)

             double precision ash, N

             k_1L_rma = 0.0005 + 0.54 * N

             return
      end
c     --------------------------------------------------------------------------------
      double precision function k_1S_rma(ash, N)

             double precision ash, N

             k_1S_rma = 0.0136 + 0.06 * ash

             return
      end
c     --------------------------------------------------------------------------------
      double precision function k_2L_rma(ash, N)

             double precision ash, N

             k_2L_rma = 0.00060

             return
       end
```

```
c       --------------------------------------------------------------------------------
        double precision function k_2S_rma(ash, N)

               double precision ash, N

               k_2S_rma = 0.00126

               return
        end
c       --------------------------------------------------------------------------------
        double precision function k_3L_rma(ash, N)

               double precision ash, N

               k_3L_rma = 0.0089 + 0.78 * N

               return
        end
c       --------------------------------------------------------------------------------
        double precision function k_3S_rma(ash, N)

               double precision ash, N

                if (ash .LT. 0.18) then
               k_3s_rma = 0.0394 - 0.21 * ash
                else
               k_3S_rma = 0.0394 - 0.21 * 0.18
                endif

               return
        end
c       --------------------------------------------------------------------------------
        double precision function k_4_rma(ash, N)

               double precision ash, N

               if (N .LE. 0.02) then
                       k_4_rma = 0.05 * N
               else
                       k_4_rma = 0.001
               endif

        return
        end
c       --------------------------------------------------------------------------------
        double precision function k_5_rma(ash, N)

               double precision ash, N
```

```fortran
                   if (N .LE. 0.005) then
                           k_5_rma = 0
                   else
                           if (N .GE. 0.02) then
                                   k_5_rma = 0.007
                           else
                                   k_5_rma = 0.007 * (100*(2*N - 0.01)/3)
                           endif
                   endif
        return
        end
c       -----------------------------------------------------------------------------
        double precision function k_6_rma()

               k_6_rma = 0.00006

        return
        end
c       -----------------------------------------------------------------------------

c ****************************************************************
c
c Subroutine simulates the soil water content in two layers,
c organic layer on top and mineral soil layer in bottom.
c
c Model presented in Linkosalo, Kolari and Pumpanen 2013.
c
c Input/output parameters: SW (SoilWater, absolute, in mm)
c                          theta (output for calc, 0 = WP and 1 = sat)
c                          prec (precipitation, in mm)
c                          ET_tot (total evapotranspiration in mm)
c
c Local parameters per layer: saturation (mm), FC (mm), WP (mm), tau (days)
c                             ET_ratio (split of ET between layers)
c
c ****************************************************************

        Subroutine two_layer_soil_water(SoilW, SWmax, theta, prec, ET_tot)

c local variables
        integer i
        real theta(2), SoilW(2), ET_tot, prec
           real FC(2), WP(2), saturation(2), ET(2)
        real tau_soil(2), ET_ratio, overflow
        real P1, P2, P_H, P_M
           real SWmax(2)

c       Soil water submodel parameters
```

```
      ET_ratio = 0.256107371
       tau_soil(1) = 0.894587365
       tau_soil(2) = 9.418136372
       saturation(1) = SWmax(1)/0.65
       saturation(2) = SWmax(2)/0.65
     wp(1) = 0
     wp(2) = 0
     FC(1) = WP(1) + SWmax(1)
     FC(2) = WP(2) + SWmax(2)

c Split evapotranspiration for the two layers
      ET(1) = ET_ratio * ET_tot
      if(ET(1) .gt. SoilW(1)) then
            ET(1) = SoilW(1)
      endif
      ET(2) = ET_tot - ET(1)

c Soilwater of organic layer from previous day over field capacity? -> overflow
      if (SoilW(1) .gt. FC(1)) then
            overflow = (SoilW(1) - FC(1)) / tau_soil(1)
            SoilW(1) = FC(1)
      else
            overflow = 0
      endif

      SoilW(1) = SoilW(1) + prec - ET(1)

c New soil water of organic layer over saturation -> immediately drainage
      if (SoilW(1) .gt. saturation(1)) then
            overflow = overflow + (SoilW(1) - saturation(1))
            SoilW(1) = saturation(1)
      endif

c          SoilW(2) = max(SoilW(2), SoilWP(2))

c Mineral soil over FC -> overflow
      if (soilW(2) .gt. FC(2)) then
            soilW(2) = soilW(2) - (soilW(2) - FC(2)) / tau_soil(2)
      endif

c Now add new water and subtract ET
      SoilW(2) = SoilW(2) + overflow - ET(2)

      if (soilW(2) .gt. saturation(2)) then
            soilW(2) = saturation(2)
c            overflow = saturation(2) - SoilW(2)
      endif

      if (SoilW(2) .lt. WP(2)) then
```

```
              ET(2) = ET(2) - (WP(2) - SoilW(2))
              soilW(2) = WP(2)
       endif

       ET_tot = ET(1) + ET(2)

       theta(1) = (soilW(1) - WP(1)) / (Saturation(1) - WP(1))
       theta(2) = (soilW(2) - WP(2)) / (Saturation(2) - WP(2))

       return
      end

c-------------------------------------------------------------------------
c
c      Evapotranspiration function based loosely on paper
c      Duursma et al. Tree Physiology 2008, but the dependency of ET on irradiation
c      modified by T. Linkosalo and fitted to Hyytiälä data in spring 2009
c
c-------------------------------------------------------------------------
       real function EvapoTranspiration (Temp, PAR, VPD, x, CO2effect,
     +          CO2ppm, REW, fDET)

c
       real ET
       real Temp, PAR, VPD, x, REW, fDET
       real beta, tau, x0, kappa, a_1, a_2, CO2ppm
       integer CO2effect

c      parameters hard-coded...

       beta = 0.016752
       tau = 14.39305
       x0 = -6.94684
       kappa = -0.000263
       a_1 = 0.0007
       a_2 = 0.0837

c      calculate S and D functions
              x = x + (Temp - x)/tau
              fS = max(0.0, x - x0)
              fD = exp(kappa * VPD)

c      calculate ET
              ET = beta * PAR * fS * fD + a_1*PAR + a_2

c Convert ET from mol/m2/d to g/m2/d
              ET = ET * 18
```

```
c Convert ET from g/m2/d to mm/d   (ET/rho and m -> mm)
            ET = ET / 1000

            fDET = 1

            if (REW .LT. 0.4) then
                    fDET = REW/0.4
                    ET = ET * fDET
            endif

            EvapoTranspiration = ET
            return
      end

c------------------------------------------------------------------------
      subroutine SoilTemperature(ST, T)

      implicit none

      real ST(2), T, adj_T(2)

c parameters for soil temperature model
      real minimum_temp(2), tau(2)
      minimum_temp(1) = -0.13
      minimum_temp(2) = 0.24
       tau(1) = 14.9
       tau(2) = 10.5

      adj_T(1) = max(T, minimum_temp(1))
      adj_T(2) = max(T, minimum_temp(2))

      ST(1) = ST(1) + (adj_T(1) - ST(1)) / tau(1)
      ST(2) = ST(2) + (adj_T(2) - ST(2)) / tau(2)

      return
      end
```

---

## Author Response (AR2)

Response to editorial comments for paper:

" Forest soil carbon stock estimates in a nationwide inventory: evaluating performance of the ROMULv and Yasso07 models in Finland" by *Lehtonen et al.* *Submitted for publication in Geoscientific Model Development journal.*

Dear Dr. Sierra,

We acknowledge editorial remarks provided to our manuscript. We have addressed those as described below (our reply is given in *italics*), note that page and line numbers refer to final version:

1. Please explain better how do you obtained the steady-state estimates. Did you run the models until a specific time? What was this time? Did you use any criteria to assess whether you reached a steady-state?

*Description of the steady-state runs were added on page 7 lines 15-18.*

2. On page 11, lines 28-32, the units are missing a time unit if they are fluxes. Are they in year^-1?

*Units were added on page 11 lines 27 – 31.*

3. Part of the discrepancy between steady-state modeled values and the observed data could be due to land-use effects. Can you discuss whether this is a possibility?

*A discussion about potential land-use effect in our soil carbon data was added on page 12 lines 21-26.*